# Genetic Relationships of *Puccinia striiformis* f. sp. *tritici* in Southwestern and Northwestern China

Bingbing Jiang,[a] Cuicui Wang,[b] Cunwu Guo,[c] Xuan Lv,[a] Wenfeng Gong,[a] Jie Chang,[d] Hongpan He,[e] Jing Feng,[f] Xianming Chen,[g] Zhanhong Ma[a]

[a]Ministry of Agriculture Key Laboratory of Pest Monitoring and Green Management, Department of Plant Pathology, China Agricultural University, Beijing, China

[b]Shandong Provincial University Laboratory for Protected Horticulture, Shandong Facility Horticulture Bioengineering Research Center, Weifang University of Science and Technology, Weifang, China

[c]College of Plant Protection, Yunnan Agricultural University, Kunming, China

[d]Yingjiang County Bureau of Agriculture and Rural Animal Husbandry Station, Yingjiang, China

[e]Wenshan Prefecture Malipo County Dong Gan Town Agricultural Integrated Service Center, Wenshan, China

[f]Gejiu City Plant Protection Plant Inspection Station, Ge Jiu, China

[g]U.S. Department of Agriculture, Agricultural Research Service, Wheat Health, Genetics, and Quality Research Unit, and Department of Plant Pathology, Washington State University, Pullman, Washington, USA

**ABSTRACT** Wheat stripe rust, caused by *Puccinia striiformis* f. sp. *tritici* (*Pst*), is a crucial disease for wheat worldwide and constantly threatens wheat production in southwestern and northwestern China, where the environment is a good fit for *Pst* oversummering and overwintering. However, the underlying genetic dynamics of spring epidemic *Pst* populations across large areas of continuous planting in the southwestern and northwestern regions are poorly understood. A total of 2,103 *Pst* isolates were sampled in the spring of 2019 from the two agroecosystems and grouped into three horizontal spatial scales (countywide, provincial, and regional subpopulations) and two vertical spatial scales that consisted of elevational and geomorphic subpopulations. A total of 776 multilocus genotypes were identified, with the highest genetic diversity found in the northern and Sichuan populations, particularly in the Ningxia and Sichuan Basins, while the lowest genetic diversity was found in the Yunnan and Guizhou populations. Multivariate discriminant analysis of principal components (DAPC) and STRUCTURE (STRUCTURE 2.3.4) analyses revealed variation in the genotypic compositions of the molecular groups on horizontal and vertical dimensions from north to south or vice versa and from low to high or vice versa, respectively. The regional neighbor-joining tree revealed three large spatial structures consisting of the southwestern, the northwestern, and the Xinjiang regions, while the Tibetan population connected the southwestern and northwestern regions. The isolates of the Sichuan Basin were scattered over the four quartiles by principal coordinate analysis, which indicated frequent genotype interchange with others. Greater genetic differentiation was observed between the southwestern and northwestern regions. Linkage equilibrium ($P \geq 0.05$) was detected on different spatial scales, suggesting that *Pst* populations are using sexual reproduction or mixed reproduction (sexual and clonal reproduction) in southwestern and northwestern China.

**IMPORTANCE** Understanding the epidemiology and population genetics of plant pathogens is crucial to formulate efficient predictions of disease outbreaks and achieve sustainable integrated disease management, especially for pathogens with migratory capability. Here, this study covers the genetic homogeneity and heterogeneity of different geographical *Pst* populations on broad to fine spatial scales from the key epidemic regions of the two agroecosystems in China, where wheat stripe rust occurs annually. We provide knowledge of the population genetics of *Pst* and reveal that, for instance, there is greater genetic diversity in northwestern China, there are close genetic relationships between Yunnan and Guizhou and between Gansu-Ningxia and Qinghai, and there are effects of

Address correspondence to Zhanhong Ma, mazh@cau.edu.cn.

The authors declare no conflict of interest.

altitude on genetic compositions, etc. All of these findings clarify the genetic relationships and expand the insights into the population dynamics and evolutionary mechanisms of *Pst* in southwestern and northwestern China, providing a theoretical basis for achieving sustainable control of wheat stripe rust in key epidemic regions.

**KEYWORDS** genetic relationships, population dynamics, broad to fine spatial scale, *Puccinia striiformis* f. sp. *tritici*, southwestern and northwestern China, spatial genetic structure, geographic subpopulations

*P*uccinia striiformis f. sp. *tritici* (*Pst*), a causal pathogen of wheat stripe rust (WSR), is distributed globally (1–7). The pathogen populations in China cannot be overlooked in worldwide epidemic studies (8–10), since WSR occurs countrywide in China annually (6, 11–16). The urediniospores of *Pst* are the most destructive form in its life cycle. Due to their strong migratory ability, the urediniospores of *Pst* can disperse thousands of kilometers through the upper air currents, which makes disease management difficult. Over the past few decades, scientists have made tremendous efforts to study *Pst* migration. They predicted the migration routes and directions among various areas of endemicity indirectly according to the wind patterns and the disease occurrence generated by national disease surveillances (11, 13). However, field surveillance requires a massive investment of manpower and physical and financial resources. In addition to the unreachable wheat planting regions in extremely mountainous areas, difficulties have led to dilemmas in disease prediction and management. Thus, it is imperative to study the genetic relationships, population genetic structures, and phylogeography on a large scale to improve the prediction and management of *Pst*.

The *Pst* populations in China possess unique features, the foremost of which is that the epidemics in the major spring epidemic regions rely on inocula from the oversummering and/or overwintering areas. *Pst* mainly oversummers in northwestern (NW) and southwestern (SW) areas at high altitudes, while overwintering occurs in SW and central regions where winter temperatures are higher (11, 13). Based on ecological conditions, Li and Zeng classified three epidemiological regions in China: the NW region, the SW region, and the Xinjiang region (11). With countless canyons and terraces, the SW and NW agroecosystems have an amazing variety of landforms, such as the Sichuan Basin, the Yunnan-Guizhou Plateau, the Qinling Mountains, and the Qinghai-Tibetan Plateau. With vast wheat production areas and an overwhelming majority of areas where *Pst* can complete its annual cycle, the SW and NW agroecosystems are the two key epidemiological regions; these consist of oversummering and/or overwintering areas and spring epidemic regions. The oversummering regions are mainly located in the two plateaus and in the Qinling Mountains, which are areas with high altitudes (11, 13). The junction areas around the Qinghai-Tibetan Plateau, the Yunnan-Guizhou Plateau, and northwestern Sichuan link the SW and NW together, connecting all of the oversummering areas (11, 13). Ali et al. inferred the Himalayas and their adjacent areas as the global origin of *Pst* based on the highest genetic diversity and distinct spatial structures (9). The Qinghai-Tibetan Plateau, the Yunnan-Guizhou Plateau, and the Qingling Mountains are on the east side of the Himalayan region. In China, it is established that Gansu Province in the northwestern region is the center of *Pst* migration, since the *Pst* population there possesses the highest diversity and disperses southward and eastward (17–19). However, Li et al. suggested the southwestern region of Yunnan Province as the new origin of *Pst* in China according to the ancestral haplotype detected in the Gansu population (20). It is necessary to make sense of *Pst* genetic dynamics between the SW and NW regions.

Phylogeographical and spatial genetic analyses of phytopathogen populations have been widely used to gain insight into potential genetic relationships (21–23). Although the genetics of *Pst* among different provinces in the SW and NW regions have been studied extensively in recent decades, previous studies have been based on small numbers of isolates from limited areas with intermittent sampling. Many researchers have focused on the genetic diversity of *Pst* populations between the SW and NW regions, and their conclusions have supported the higher genetic diversity in NW populations, which was highest in

Longnan of Gansu (17, 18, 24, 25). However, others have mainly explored the genetic differentiation and dynamics between the SW and NW regions and found various amounts of gene flow among the junction areas around Gansu, Ningxia, Qinghai, Shaanxi, and Sichuan (19, 26–28). Meanwhile, Xinjiang and Tibetan populations are relatively isolated from inland populations (29–31).

Nevertheless, there are still many questions to answer. For instance, in spring epidemics, what is the genetic connection and differentiation between the SW and NW *Pst* populations? What are the functions of regional *Pst* subpopulations, for example, in northwestern Sichuan, southeastern Tibet, central Yunnan, and western Guizhou? In addition, the effects of diverse cropping systems in these regions, such as overlapping sowing and harvesting times, complex varieties, and variable climates, all lead to indistinct genetic relationships and migrations patterns (6, 11, 13, 15, 16). In particular, the phylogeography of *Pst* populations across continuous regions over large areas is poorly understood, and the understanding of genetic homogeneity and heterogeneity is still lacking. Hence, it is vital to study the genetics of *Pst* in the SW and NW regions to improve the management of WSR in China. Therefore, we obtained *Pst* isolates extensively and intensively across the continuous wheat planting regions in SW and NW China and explored their genetic relationships by using simple sequence repeat (SSR) markers. The objectives of this study were to improve our understanding of (i) the genetic structure and spatial distribution among the various geographical subpopulations, (ii) the genetic diversity and differentiation within and among different spatial subpopulations, and (iii) the potential reproduction mode of *Pst* in SW and NW China.

## RESULTS

**SSR markers.** All 12 pairs of SSR markers were polymorphic among the *Pst* isolates (see Table S1), and the genotype accumulation curves showed that 97% (776 of 800) of the multilocus genotypes (MLGs) could be identified by 11 pairs of markers, indicating that 12 loci were sufficient for further genetic analysis in the present study (see Fig. S1). The numbers of alleles in the 12 SSR loci ranged from 1.400 (RJN8) to 4.700 (CPS34), with an average of 2.567 alleles per locus. The ranges of observed heterozygosity ($H_o$), expected heterozygosity ($H_e$), and polymorphism information content values across the 12 pairs of markers were from 0.048 to 0.404, 0.041 to 0.424, and 0.070 to 0.701, respectively.

**Genetic diversity.** A total of 776 genotypes were detected from the total of 2,103 *Pst* isolates from SW and NW China in spring 2019 (see Table S4). Provincial subpopulation abbreviations were as follows (see Table S2): CQ (Chongqing), SC (Sichuan), GZ (Guizhou), YN (Yunnan), XZ (Tibet), NX (Ningxia), GS (Gansu), QH (Qinghai), SX (Shaanxi), and XJ (Xinjiang). Regional subpopulation abbreviations were as follows: SB (Sichuan Basin), SN (northern Sichuan), SSW (southwestern Sichuan), SNW (northwestern Sichuan), GW (western Guizhou), YNE (northeastern Yunnan), YC (central Yunnan), YSE (southeastern Yunnan), YW (western Yunnan), ZSE (southeastern Tibet), and SG (Shaanxi and Gansu). Abbreviations for the provincial subpopulations NX, QH, and XJ were the same as those used for the regional subpopulations. Due to fewer than three sampling villages or sites, the isolates from Xinjiang Province were excluded during the genetic diversity analysis at the provincial and regional levels (groups P and R) but included as a part of groups E and L, and the Shaanxi population was excluded at the provincial level but included as a part of groups R, E, and L. At the regional level (group R), the *Pst* isolates from Qianyang County of Central Shaanxi and Zhuanglang County of Longdong District in Gansu Province were grouped into one subpopulation, abbreviated as SG (Shaanxi and Gansu), based on the close genetic relationships in these regions according to Wang (27).

At the provincial level (group P) (Table 1), the genotypic diversities (MLG ratio, eMLG, $H$, $G$, $\lambda$, and corrected $\lambda$) of QH (0.607, 9.910, 3.936, 29.694, 0.966, and 0.974, respectively), SC (0.388, 10.230, 4.895, 57.008, 0.982, and 0.983), and NX (0.579, 9.941, 4.021, 33.677, 0.970, and 0.977) were relatively high; and the gene diversities ($H_o$, $H_e$, $N_a$, and $N_e$) in Tibet (0.315, 0.318, 3.250, and 1.638), SC (0.226, 0.309, 3.833, and 1.596), CQ (0.262, 0.306, 3.250, and 1.619), and QH (0.333, 0.296, 2.583, and 1.502) were relatively high, where the total genetic diversity (genotypic and gene diversity) was lowest in GZ (genotypic diversity: 0.380, 7.756, 2.791, 8.422, 0.881, and 0.892; gene diversity: 0.200, 0.217, 2.583, and 1.331) and YN

**TABLE 1** Genetic diversity statistics for subpopulations at the provincial level (group P) based on 12 SSR loci[a]

| Group P | N | Genotypic diversity | | | Index of MLG diversity | | | | | Gene diversity | | | | | |
| | | MLG | MLG ratio | eMLG | H | G | λ | Corrected λ | E.5 | $H_o$ | $H_e$ | $N_a$ | $N_e$ | $A_p$ | PL |
|---|---|---|---|---|---|---|---|---|---|---|---|---|---|---|---|
| CQ | 182 | 77 | 0.423 | 9.007 | 3.657 | 17.751 | 0.944 | 0.949 | 0.444 | 0.262 | 0.306 | 3.250 | 1.619 | 0.083 | 1.000 |
| SC | 681 | 264 | 0.388 | 10.230 | 4.895 | 57.008 | 0.982 | 0.983 | 0.423 | 0.226 | 0.309 | 3.833 | 1.596 | 0.417 | 0.917 |
| GZ | 79 | 30 | 0.380 | 7.756 | 2.791 | 8.422 | 0.881 | 0.892 | 0.485 | 0.200 | 0.217 | 2.583 | 1.331 | 0.000 | 1.000 |
| YN | 537 | 180 | 0.335 | 9.575 | 4.329 | 28.453 | 0.965 | 0.967 | 0.367 | 0.240 | 0.256 | 2.917 | 1.499 | 0.083 | 0.917 |
| XZ | 318 | 156 | 0.491 | 9.719 | 4.409 | 27.450 | 0.964 | 0.967 | 0.326 | 0.315 | 0.318 | 3.250 | 1.638 | 0.083 | 1.000 |
| NX | 140 | 81 | 0.579 | 9.941 | 4.021 | 33.677 | 0.970 | 0.977 | 0.597 | 0.270 | 0.267 | 2.333 | 1.426 | 0.000 | 0.917 |
| GS | 25 | 14 | 0.560 | 8.471 | 2.512 | 10.965 | 0.909 | 0.947 | 0.880 | 0.200 | 0.298 | 2.000 | 1.559 | 0.000 | 0.750 |
| QH | 117 | 71 | 0.607 | 9.910 | 3.936 | 29.694 | 0.966 | 0.974 | 0.572 | 0.333 | 0.296 | 2.583 | 1.502 | 0.000 | 0.917 |
| Grand total/mean | 2,103 | 776 | 0.369 | 10.416 | 5.718 | 78.771 | 0.987 | 0.987 | 0.256 | 0.270 | 0.332 | 2.567 | 1.486 | 0.067 | 0.833 |

[a]N, number of Pst isolates from each province; MLG, number of multilocus genotypes or genotypic richness observed; MLG ratio, genetic diversity of MLGs; eMLG, number of expected MLGs based on rarefaction correction; H, Shannon-Winner index of MLG diversity; G, Stoddart and Taylor's index of MLG diversity; λ and corrected λ, Simpson's index; E.5, evenness, a measure of distribution of genotype abundance in a sample; $H_o$, observed heterozygosity; $H_e$, expected heterozygosity; $N_a$, number of observed alleles; $N_e$, effective number of alleles; $A_p$, number of private alleles; PL, percentage of polymorphic loci. Provincial subpopulation abbreviations: CQ, Chongqing; SC, Sichuan; GZ, Guizhou; YN, Yunnan; XZ, Tibet; NX, Ningxia; GS, Gansu; QH, Qinghai.

(genotypic diversity: 9.575, 4.329, 28.453, 0.965, and 0.967; gene diversity: 0.240, 0.256, 2.917, and 1.499). The subpopulations of CQ, GZ, and XZ had 100% PL (percentage of polymorphic loci). At the regional level (group R) (Table 2), the genotypic diversities of QH, NX, and SN (0.402, 9.742, 3.972, 32.582, 0.969, and 0.974) and SB (0.439, 9.708, 4.365, 29.970, 0.967, and 0.969) were relatively high, and the gene diversities of ZSE, QH, and SB (0.252, 0.321, 3.917, and 1.617) and NX were relatively high, where the total genetic diversity was lowest in GW. The subpopulations of SB, GW, and ZSE had 100% PL. At the elevational (group E) level (Table 3), the genotypic diversities of the subpopulations 1.4K to 1.6K (1.4-1.6K; 0.484, 9.395, 4.475, 55.082, 0.982, and 0.985), 2.0-2.2K (0.604, 9.382, 4.417, 47.751, 0.979, and 0.984), 2.4-2.6K (0.564, 8.003, 2.810, 11.436, 0.913, and 0.937), and 2.6-2.8K (0.660, 9.404, 4.015, 41.280, 0.976, and 0.985) were relatively high, while the gene diversities of the subpopulations 2.0-2.2K (0.270, 0.292, 3.083, and 1.573), 2.4-2.6K (0.392, 0.310, 2.250, and 1.644), and 3.0-3.2K (0.331, 0.369, 2.500, and 1.695) were relatively high; thus, the lowest total genetic diversity was found in the range of 0.6-0.8K (genotypic diversity: 0.314, 5.429, 1.901, 4.804, 0.792, and 0.815, gene diversity: 0.198, 0.267, 2.250, and 1.457), with private alleles detected in the subpopulations 0.2-0.4K, 0.4-0.6K, 0.8-1.0K, and 1.8-2.0K, and the highest was in the subpopulation 0.4-0.6K (0.333). The isolates from the 0.2-0.4K, 1.4-1.6K, 2.0-2.2K, and 2.6-2.8K regions had 100% polymorphic loci. At the geomorphic (group L) level (Table 4), while the genotypic diversity and gene diversity were highest in the EHM (extremely high mountain; genotypic diversity: 0.562, 141.000, 4.557, 54.264, 0.982, and 0.985, gene diversity: 0.309, 0.326, 3.500,

**TABLE 2** Genetic diversity statistics for subpopulations at the regional level (group R) based on 12 SSR loci[a]

| Group R | N | Genotypic diversity | | | Index of MLG diversity | | | | | Gene diversity | | | | | |
| | | MLG | MLG ratio | eMLG | H | G | λ | Corrected λ | E.5 | $H_o$ | $H_e$ | $N_a$ | $N_e$ | $A_p$ | PL |
|---|---|---|---|---|---|---|---|---|---|---|---|---|---|---|---|
| SB | 355 | 156 | 0.439 | 9.708 | 4.365 | 29.970 | 0.967 | 0.969 | 0.373 | 0.252 | 0.321 | 3.917 | 1.617 | 0.500 | 1.000 |
| SN | 224 | 90 | 0.402 | 9.742 | 3.972 | 32.582 | 0.969 | 0.974 | 0.611 | 0.197 | 0.281 | 2.667 | 1.467 | 0.000 | 0.917 |
| SSW | 213 | 78 | 0.366 | 9.258 | 3.745 | 21.270 | 0.953 | 0.957 | 0.471 | 0.244 | 0.252 | 2.667 | 1.572 | 0.083 | 0.917 |
| SNW | 71 | 39 | 0.549 | 8.574 | 3.148 | 11.031 | 0.909 | 0.922 | 0.505 | 0.227 | 0.280 | 2.500 | 1.506 | 0.000 | 0.917 |
| GW | 79 | 30 | 0.380 | 7.756 | 2.791 | 8.422 | 0.881 | 0.893 | 0.485 | 0.200 | 0.217 | 2.583 | 1.331 | 0.000 | 1.000 |
| YNE | 221 | 93 | 0.421 | 9.533 | 3.949 | 26.733 | 0.963 | 0.967 | 0.506 | 0.245 | 0.280 | 2.583 | 1.529 | 0.000 | 0.917 |
| YC | 168 | 74 | 0.440 | 8.954 | 3.601 | 18.233 | 0.945 | 0.951 | 0.483 | 0.232 | 0.218 | 2.583 | 1.400 | 0.000 | 0.917 |
| YSE | 56 | 34 | 0.607 | 9.275 | 3.229 | 17.231 | 0.942 | 0.959 | 0.669 | 0.223 | 0.224 | 2.500 | 1.455 | 0.000 | 0.833 |
| YW | 92 | 45 | 0.489 | 8.889 | 3.316 | 15.389 | 0.935 | 0.945 | 0.542 | 0.252 | 0.244 | 2.417 | 1.552 | 0.000 | 0.917 |
| ZSE | 318 | 156 | 0.491 | 9.719 | 4.409 | 27.450 | 0.964 | 0.967 | 0.326 | 0.315 | 0.318 | 3.250 | 1.638 | 0.083 | 1.000 |
| NX | 140 | 81 | 0.579 | 9.941 | 4.021 | 33.677 | 0.970 | 0.977 | 0.597 | 0.270 | 0.267 | 2.333 | 1.426 | 0.000 | 0.917 |
| SG | 36 | 18 | 0.500 | 9.697 | 2.702 | 12.462 | 0.920 | 0.946 | 0.824 | 0.248 | 0.298 | 2.333 | 1.555 | 0.000 | 0.833 |
| QH | 117 | 71 | 0.607 | 9.910 | 3.936 | 29.694 | 0.966 | 0.975 | 0.572 | 0.338 | 0.296 | 2.583 | 1.502 | 0.000 | 0.917 |
| Grand total/mean | 2,103 | 776 | 0.369 | 10.416 | 5.718 | 78.771 | 0.987 | 0.988 | 0.256 | 0.250 | 0.332 | 2.500 | 1.483 | 0.044 | 0.856 |

[a]See Table 1, footnote a. Regional subpopulation abbreviations: SB, Sichuan Basin; SN, northern Sichuan; SSW, southwestern Sichuan; SNW, northwestern Sichuan; GW, western Guizhou; YNE, northeastern Yunnan; YC, central Yunnan; YSE, southeastern Yunnan; YW, western Yunnan; ZSE, southeastern Tibet; SG, Shaanxi and Gansu; QH, Qinghai.

**TABLE 3** Genetic diversity statistics for subpopulations at the elevational level (group E) based on 12 SSR loci[a]

| Group E | N | Genotypic diversity | | | Index of MLG diversity | | | | | Gene diversity | | | | | |
|---|---|---|---|---|---|---|---|---|---|---|---|---|---|---|---|
| | | MLG | MLG ratio | eMLG | H | G | λ | Corrected λ | E.5 | $H_o$ | $H_e$ | $N_a$ | $N_e$ | $A_p$ | PL |
| 0.0-0.2K | 10 | 9 | 0.900 | 9.000 | 2.164 | 8.333 | 0.880 | 0.978 | 0.952 | 0.308 | 0.248 | 1.667 | 1.461 | 0.000 | 0.583 |
| 0.2-0.4K | 169 | 86 | 0.509 | 8.867 | 3.979 | 26.470 | 0.962 | 0.968 | 0.486 | 0.258 | 0.307 | 3.250 | 1.627 | 0.083 | 1.000 |
| 0.4-0.6K | 378 | 152 | 0.402 | 9.144 | 4.409 | 40.661 | 0.975 | 0.978 | 0.488 | 0.220 | 0.308 | 3.667 | 1.543 | 0.333 | 91.670 |
| 0.6-0.8K | 35 | 11 | 0.314 | 5.429 | 1.901 | 4.804 | 0.792 | 0.815 | 0.668 | 0.198 | 0.267 | 2.250 | 1.457 | 0.000 | 0.833 |
| 0.8-1.0K | 27 | 17 | 0.630 | 7.921 | 2.602 | 10.268 | 0.903 | 0.937 | 0.742 | 0.272 | 0.246 | 2.083 | 1.513 | 0.083 | 0.583 |
| 1.0-1.2K | 95 | 31 | 0.326 | 6.585 | 2.615 | 6.895 | 0.855 | 0.864 | 0.465 | 0.262 | 0.196 | 2.083 | 1.518 | 0.000 | 0.500 |
| 1.2-1.4K | 117 | 49 | 0.419 | 7.321 | 3.069 | 10.073 | 0.901 | 0.908 | 0.442 | 0.229 | 0.281 | 2.583 | 1.509 | 0.000 | 0.917 |
| 1.4-1.6K | 277 | 134 | 0.484 | 9.395 | 4.475 | 55.082 | 0.982 | 0.985 | 0.623 | 0.222 | 0.273 | 3.000 | 1.483 | 0.000 | 1.000 |
| 1.6-1.8K | 193 | 90 | 0.466 | 9.079 | 4.069 | 34.845 | 0.971 | 0.976 | 0.589 | 0.240 | 0.226 | 2.583 | 1.420 | 0.000 | 0.917 |
| 1.8-2.0K | 177 | 91 | 0.514 | 9.070 | 4.066 | 34.165 | 0.971 | 0.976 | 0.579 | 0.239 | 0.235 | 2.583 | 1.516 | 0.083 | 0.917 |
| 2.0-2.2K | 192 | 116 | 0.604 | 9.382 | 4.417 | 47.751 | 0.979 | 0.984 | 0.571 | 0.270 | 0.292 | 3.083 | 1.573 | 0.000 | 1.000 |
| 2.2-2.4K | 39 | 22 | 0.564 | 8.003 | 2.810 | 11.436 | 0.913 | 0.937 | 0.669 | 0.387 | 0.311 | 2.500 | 1.625 | 0.000 | 0.833 |
| 2.4-2.6K | 20 | 15 | 0.750 | 8.684 | 2.623 | 12.500 | 0.920 | 0.968 | 0.900 | 0.392 | 0.310 | 2.250 | 1.644 | 0.000 | 0.917 |
| 2.6-2.8K | 103 | 68 | 0.660 | 9.404 | 4.015 | 41.280 | 0.976 | 0.985 | 0.740 | 0.319 | 0.289 | 2.667 | 1.526 | 0.000 | 1.000 |
| 2.8-3.0K | 211 | 118 | 0.559 | 8.535 | 4.090 | 18.164 | 0.945 | 0.949 | 0.292 | 0.296 | 0.290 | 2.667 | 1.486 | 0.000 | 0.917 |
| 3.0-3.2K | 60 | 22 | 0.367 | 7.694 | 2.780 | 12.587 | 0.921 | 0.936 | 0.767 | 0.331 | 0.369 | 2.500 | 1.695 | 0.000 | 0.917 |
| Grand total/mean | 2,103 | 776 | 0.369 | 9.515 | 5.718 | 78.771 | 0.987 | 0.988 | 0.256 | 0.278 | 0.278 | 2.589 | 1.537 | 0.036 | 0.860 |

[a]See Table 1, footnote a.

and 1.657) and HM (high mountain; genotypic diversity: 0.382, 131.169, 4.805, 52.982, 0.981, and 0.983, gene diversity: 0.270, 0.334, 2.833, and 1.598) subpopulations, the lowest were in the plains (genotypic diversity: 0.332, 85.797, 3.623, 15.673, 0.936, and 0.940, gene diversity: 0.240, 0.256, 3.167, and 1.549) and lowland (genotypic diversity: 0.408, 131.076, 4.710, 43.192, 0.977, and 0.979) subpopulations. The isolates from the lowland, plains, and EHM subpopulations contained private alleles, with the largest amount being found in the lowland subpopulation (0.417). The isolates from the lowland, highland, and EHM subpopulations had 100% polymorphic loci.

On the horizontal spatial scale, the shared and dominant MLGs mainly occurred in the southwestern subpopulations despite the provincial and regional spatial levels, and the proportions were approximately 50% (Fig. 1B and C). The SC and YN provinces or regional subpopulations SB, SSW, YNE, and SN showed relatively higher numbers and frequencies of shared MLGs (Tables 5 and 6). Except for MLG 114 and MLG 80, which were found in the Tibet and SNW subpopulations, the other southwestern subpopulations mainly contained the dominant MLGs 524, 633, 632, 86, 707, and 485. The northwestern subpopulations were generally composed of private MLGs, although QH contained some shared MLGs. On the vertical spatial scale, each elevational and geomorphic subpopulation contained nearly equal amounts of shared and private MLGs (Fig. 1D and E), and the subpopulations 0.4-0.6K, 1.4-1.6K, 1.6-1.8K, 1.8-2.0K, highland, and lowland showed a higher number and frequency of shared MLGs (Tables 7 and 8). Each subpopulation included dominant MLGs but had diverse mixabilities. In general, as the altitude increased, the singleness of dominant MLGs increased, while at an elevation of 2,800 m, only MLGs 114 and 707 prevailed.

Across all four spatial groups, the genetic diversity declined from north to south and from high to low altitude, as well as the variation in genotypic structures. In general,

**TABLE 4** Genetic diversity statistics for subpopulations at the geomorphic level (group L) based on 12 SSR loci[a]

| Group L | N | Genotypic diversity | | | Index of MLG diversity | | | | | Gene diversity | | | | | |
|---|---|---|---|---|---|---|---|---|---|---|---|---|---|---|---|
| | | MLG | MLG ratio | eMLG | H | G | λ | Corrected λ | E.5 | $H_o$ | $H_e$ | $N_a$ | $N_e$ | $A_p$ | PL |
| Lowland | 557 | 227 | 0.408 | 131.076 | 4.710 | 43.192 | 0.977 | 0.979 | 0.383 | 0.234 | 0.315 | 3.917 | 1.582 | 0.417 | 1.000 |
| Plain | 274 | 91 | 0.332 | 85.797 | 3.623 | 15.673 | 0.936 | 0.940 | 0.402 | 0.240 | 0.256 | 3.167 | 1.549 | 0.083 | 0.917 |
| Highland | 374 | 193 | 0.516 | 143.924 | 4.675 | 36.200 | 0.972 | 0.975 | 0.331 | 0.242 | 0.301 | 3.417 | 1.592 | 0.000 | 1.000 |
| HM | 647 | 247 | 0.382 | 131.169 | 4.805 | 52.982 | 0.981 | 0.983 | 0.429 | 0.270 | 0.334 | 2.833 | 1.598 | 0.000 | 0.917 |
| EHM | 251 | 141 | 0.562 | 141.000 | 4.557 | 54.264 | 0.982 | 0.985 | 0.565 | 0.309 | 0.326 | 3.500 | 1.657 | 0.167 | 1.000 |
| Grand total/mean | 2,103 | 776 | 0.369 | 163.840 | 5.718 | 78.771 | 0.987 | 0.988 | 0.256 | 0.259 | 0.306 | 3.367 | 1.596 | 0.133 | 0.967 |

[a]See Table 1, footnote a. Geomorphic subpopulation abbreviations: HM, high mountain; EHM, extremely high mountain.

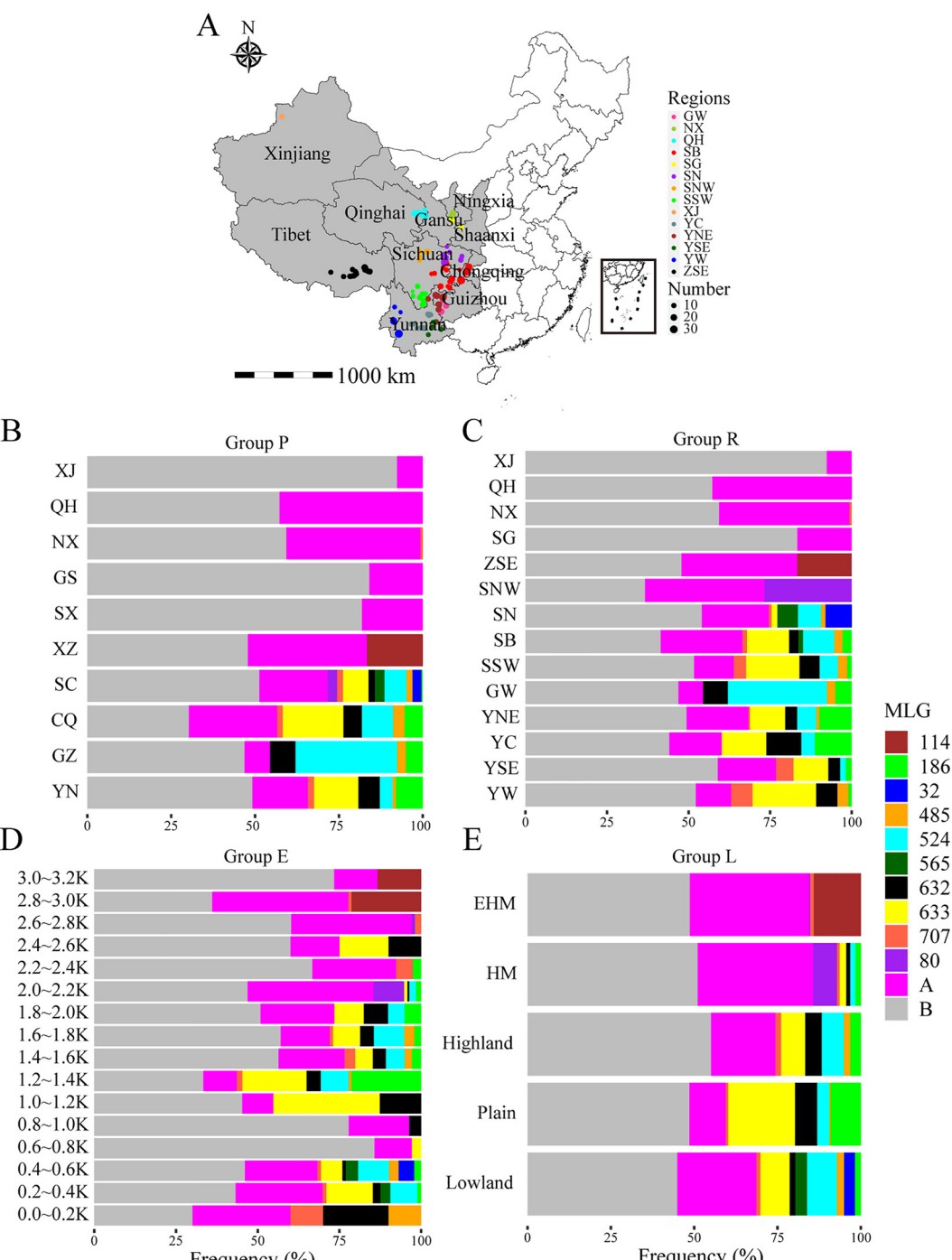

**FIG 1** Sampling locations for 2,103 *Pst* samples in southwestern (SW) and northwestern (NW) China and compositions of MLGs in the four spatial subpopulations. (A) The color represents the region, and the size of the circle represents the number of samples. Provincial subpopulation abbreviations: CQ, Chongqing; SC, Sichuan; GZ, Guizhou; YN, Yunnan; XZ, Tibet; NX, Ningxia; GS, Gansu; QH, Qinghai; SX, Shaanxi; XJ, Xinjiang. Regional subpopulation abbreviations: SB, Sichuan Basin; SN, northern Sichuan; SSW, southwestern Sichuan; SNW, northwestern Sichuan; GW, western Guizhou; YNE, northeastern Yunnan; YC, central Yunnan; YSE, southeastern Yunnan; YW, western Yunnan; ZSE, southeastern Tibet; SG, Shaanxi and Gansu. Geomorphic subpopulation abbreviations: EHM, extremely high mountain; HM, high mountain. (B to E) Compositions of MLGs in subpopulations under provincial, regional, elevational, and geomorphic spatial scales. MLGs 114, 186, 32, 485, 524, 565, 632, 633, 707, and 80 represented the dominant MLGs. A, Other nondominant shared MLGs; B, private MLGs in one subpopulation. The *x* coordinate indicates the frequency.

**TABLE 5** Shared MLGs at the provincial level

| Group P | MLG[a] | CQ | SC | GZ | YN | XZ | NX | GS | QH | SX | XJ | Total |
|---|---|---|---|---|---|---|---|---|---|---|---|---|
| Among | No. | 18 | 58 | 24 | 55 | 2 | 12 | 5 | 3 | 0 | 0 | 1,054 |
| | Freq. | 109 | 281 | 73 | 330 | 11 | 45 | 7 | 4 | 0 | 0 | 860 |
| Within | No. | 11 | 67 | 0 | 36 | 41 | 13 | 5 | 18 | 2 | 1 | 194 |
| | Freq. | 25 | 261 | 0 | 118 | 194 | 39 | 14 | 63 | 9 | 12 | 735 |

[a]MLG represents the number of multilocus genotypes or genotypic richness observed. No., number of MLGs; Freq. (frequency), number of individuals that were in the same MLG.

the dominant and shared MLGs existed in the southwestern subpopulation, while the private MLGs existed in the northwestern subpopulation, and vertically, the subpopulations with extremely high altitudes possessed the purer, dominant MLGs. All the results indicate that genetic diversity is affected by geography.

**Population subdivision.** The results generated from the DAPC (multivariate discriminant analysis of principal components) with a nonparametric model and STRUCTURE 2.3.4 analysis with a Bayesian model displayed similar patterns of genetic structure in genotypic molecular groups (MGs) and revealed that the best number of putative clusters (K) to discriminate the 2,103 *Pst* isolates into different MGs was 2 (Fig. 2; see also Fig. S2). When K = 2, the brown cluster (MG1) was constructed by isolates from the majority of YN and GZ isolates and half of the CQ and SC isolates, mainly from the SSW region, half of the SB region, and a small part of the SNW region. The green cluster (MG2) was mainly formed by the isolates from Tibet, the northwestern subpopulations except for XJ, half of CQ and SC, most of SN and SNW, and half of SB, with a few isolates from YN and GZ (Fig. 2B). When K = 3, the three MGs displayed little overlap (Fig. 2C). The distributions of brown (MG1) and green (MG2) clusters were basically consistent with K = 2, and the new blue cluster (MG3) occurred in southwestern regions, particularly in CQ and SC (Fig. 2D). When K = 4, the discrimination displayed partial overlapping and were indistinguishable, and the previous green cluster even split into two clusters, as shown by the orange and green colors (see Fig. S2).

Furthermore, to depict the horizontal and vertical genetic connectivity over a large area, pie charts were constructed from the different genotypic proportions in groups P, R, E, and L. Due to the similarity of the bar plots generated by the two different cluster methods, we only displayed the results of the nonparametric model (Fig. 3). When K = 2, in group P, the genetic variation within the green cluster (MG2), from every sampled province, showed a decline from the northwestern toward the southwestern regions, whereas the genetic variation among the brown cluster (MG1) declined from the southwestern toward the northwestern regions (Fig. 3A and B). This phenomenon was more pronounced in group R, the decrease in genetic variation within MG2 started in the Qinghai-Tibet Plateau and moved southeastward, while it increased among MG1 from west to east and north to south (Fig. 3C). For the vertical distribution, when K = 2 and the elevation was above 1,400 m, the genetic variation within MG1 decreased, while it increased among MG2 as the elevation increased (Fig. 3G). The subpopulations under 2,000 m are mainly composed of MG1; in contrast, genetic variation within MG2 increased when the elevation was above 2,000 m.

Based on Nei's genetic distance, the 10 provincial subpopulations were classified into four clusters when the distance was 0.05 (Fig. 4A). The first cluster included all subpopulations from the southwestern region, and the second cluster consisted of the northwestern subpopulations of QH, NX, and GS; the third and fourth clusters were constructed by SX and XJ, respectively. The regional neighbor-joining tree showed a distinct structure with six clusters based on a distance of 0.025 (Fig. 4B). The first cluster comprised the SNW, SN, SB, and

**TABLE 6** Shared MLGs at the regional level

| Group R | MLG[a] | SB | SN | SNW | SSW | GW | YNE | YC | YSE | YW | ZSE | NX | SG | QH | XJ | Total |
|---|---|---|---|---|---|---|---|---|---|---|---|---|---|---|---|---|
| Among | No. | 36 | 26 | 5 | 40 | 24 | 38 | 44 | 24 | 34 | 2 | 12 | 5 | 3 | 0 | 102 |
| | Freq. | 178 | 88 | 8 | 157 | 73 | 133 | 135 | 46 | 80 | 11 | 45 | 7 | 4 | 0 | 965 |
| Within | No. | 31 | 18 | 8 | 12 | 0 | 13 | 3 | 0 | 1 | 41 | 13 | 7 | 18 | 1 | 166 |
| | Freq. | 88 | 90 | 37 | 30 | 0 | 46 | 6 | 0 | 2 | 194 | 39 | 23 | 63 | 12 | 630 |

[a]See Table 5, footnote *a*.

**TABLE 7** Shared MLGs at the elevational level

| Group E | MLG[a] | 0.0-0.2K | 0.2-0.4K | 0.4-0.6K | 0.6-0.8K | 0.8-1.0K | 1.0-1.2K | 1.2-1.4K | 1.4-1.6K | 1.6-1.8K | 1.8-2.0K | 2.0-2.2K | 2.2-2.4K | 2.4-2.6K | 2.6-2.8K | 2.8-3.0K | 3.0-3.2K | Total |
|---|---|---|---|---|---|---|---|---|---|---|---|---|---|---|---|---|---|---|
| Among | No. | 6 | 26 | 40 | 4 | 10 | 21 | 36 | 72 | 52 | 42 | 24 | 12 | 10 | 15 | 12 | 6 | 133 |
| | Freq. | 7 | 90 | 179 | 13 | 13 | 84 | 103 | 207 | 143 | 114 | 72 | 29 | 13 | 20 | 61 | 25 | 1,173 |
| Within | No. | 0 | 15 | 28 | 3 | 2 | 1 | 1 | 6 | 9 | 9 | 18 | 0 | 2 | 15 | 18 | 8 | 135 |
| | Freq. | 0 | 34 | 115 | 18 | 9 | 2 | 2 | 14 | 21 | 23 | 46 | 0 | 4 | 45 | 62 | 27 | 422 |

[a]See Table 5, footnote a.

GW subpopulations; the second cluster included all the YN and SSW subpopulations; the third cluster was formed by the QH and NX subpopulations; and the fourth, fifth, and sixth clusters were represented by the ZES, SG, and XJ subpopulations, respectively. Here, we noticed that the Tibetan subpopulation was closer to the YN and QH subpopulations and connected the two areas. In group E (Fig. 4C), three clusters were identified based on a distance of 0.04: the first cluster consisted of the subpopulations with elevations from 2,000 to 2,600 m, the second cluster included the subpopulations with elevations above 2,600 m, and the third cluster contained the remaining subpopulations below 2,000 m. In group L (Fig. 4D), two clusters were established by the five landforms based on a distance of 0.32; the HM and EHM subpopulations formed the first cluster, while the lowland, plain, and highland subpopulations formed the second cluster.

Principal coordinate analysis (PCoA) showed that horizontal and vertical coordinates accounted for 22.92 and 15.35% of the total genetic variance, respectively. At the provincial level (Fig. 4E), the isolates from the CQ subpopulation were mostly located in the first, second, and third quartiles; isolates of the GZ subpopulation were mostly located in the third and fourth quartiles; isolates of the YN subpopulation were mostly located in the first, third, and fourth quartiles; isolates of the Tibetan subpopulation were mostly located in the first and second quartiles; isolates from the NX, QH, and GS subpopulations were mostly located in the first and second quartiles; and isolates from the XJ subpopulation were located in the first quartile. Isolates of the SC subpopulation showed a scattered distribution. At the regional level (Fig. 4F), the isolates of the SSW subpopulation were mostly distributed in the first and fourth quartiles; the isolates of the SN and SNW subpopulations were mainly located in the second, third, and fourth quartiles; the isolates of the GW subpopulation were distributed in the third and fourth quartiles; the isolates of the ZSE subpopulation were located in the first and second quartiles; the isolates of the YNE and YC subpopulations were mainly distributed in the first, third and fourth quartiles, while the isolates of the YSE and YW subpopulations were mainly located in the first and fourth quartiles. For the northwest, isolates of the NX, SG, and QH subpopulations were mostly located in the first and second quartiles; isolates of the XJ subpopulation were located in the first quartile; and isolates of the SB subpopulation were scattered. In the elevational and geomorphic groups (Fig. 4G and H), the lowland and highland subpopulations had scattered distributions, while the isolates of the plain subpopulation were located in the first and fourth quartiles, and the isolates of the EHM subpopulation were mostly distributed in the third and fourth quartiles.

The pairwise comparisons of G-statistics based on the 2,103 *Pst* isolates among the four spatial groups revealed genetic differentiation between the SW and NW subpopulations (Fig. 5; see also Table S3). In group P (Fig. 5A; see also Table S3A), the highest genetic differentiation was between the XJ subpopulation and others, particularly compared to the SX subpopulation

**TABLE 8** Shared MLGs at the geomorphic level

| Group L | MLG[a] | Lowland | Plain | Highland | HM | EHM | Total |
|---|---|---|---|---|---|---|---|
| Among | No. | 44 | 50 | 80 | 31 | 8 | 90 |
| | Freq. | 222 | 201 | 397 | 94 | 14 | 928 |
| Within | No. | 51 | 11 | 42 | 23 | 51 | 178 |
| | Freq. | 203 | 43 | 125 | 70 | 226 | 667 |

[a]See Table 5, footnote a.

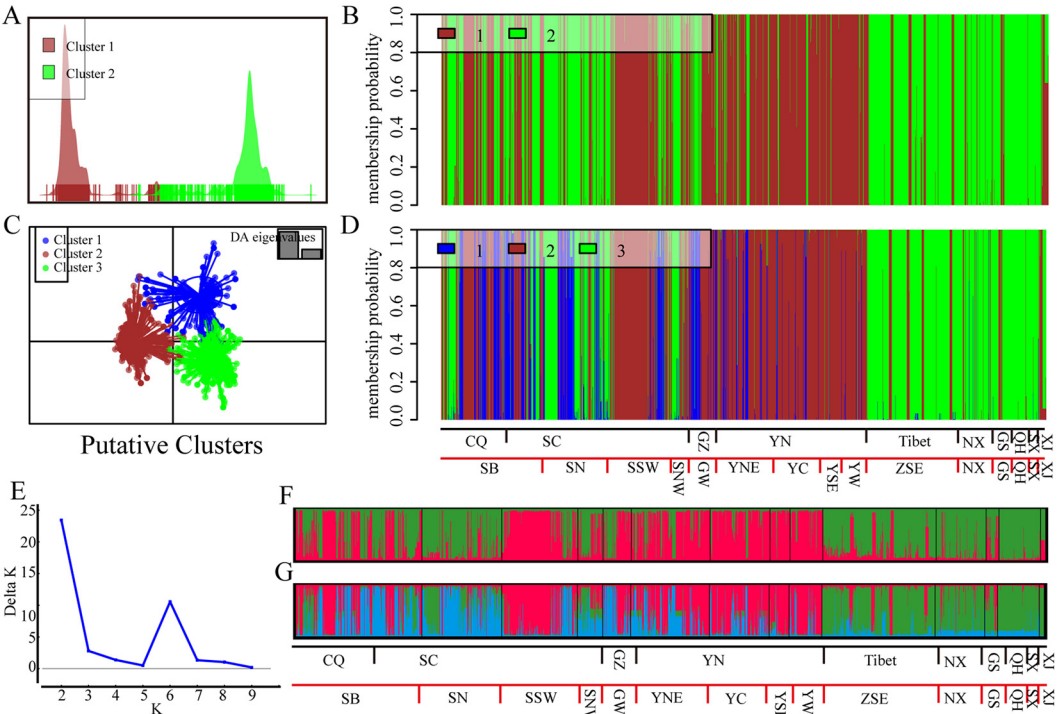

**FIG 2** Cluster analysis implemented in the nonparametric model method of DAPC (A to D) and Bayesian model-based method of the STRUCTURE software (E to G) assigned 2,103 *Pst* isolates into different molecular groups (MGs). (A) In the DAPC analysis, when the putative cluster was 2, all 2,103 *Pst* isolates could be separated. (B) A bar plot displays the membership coefficient for individuals in the two MGs. (C) In the DAPC analysis, when the putative cluster was 3, there was little overlap in the discrimination. (D) Membership coefficient for individuals in the three MGs. (E) STRUCTURE estimated that the best K was 2. (F and G) Membership coefficients for individuals in the two and three MGs classified by STRUCTURE.

($G_{st}$ = 0.462), followed by that between the SX subpopulation and others, compared to the GZ subpopulation as the maximum ($G_{st}$ = 0.305), secondary compared to the YN subpopulation ($G_{st}$ = 0.23). There were certain degrees of differentiation between the other subpopulations of the northwestern and southwestern regions. For instance, the GS subpopulation compared to the GZ subpopulation ($G_{st}$ = 0.234), the QH subpopulation compared with the GZ subpopulation ($G_{st}$ = 0.226) and the QH subpopulation compared to the YN subpopulation ($G_{st}$ = 0.225), and the SX subpopulation compared to the XZ subpopulation ($G_{st}$ = 0.226). For the southwestern subpopulations, the genetic differentiations were negligible; only moderate genetic differentiation existed between the XZ subpopulation and others, except it was the lowest compared to the SC ($G_{st}$ = 0.049) and CQ ($G_{st}$ = 0.066) subpopulations and the highest compared to the XJ ($G_{st}$ = 0.285) and SX ($G_{st}$ = 0.226) subpopulations. All $P$ values were significant ($P$ = 0.001). In group R (Fig. 5B; see also Table S3B); there was slight genetic differentiation among the northwestern subpopulations. The greatest genetic differentiations were still between the XJ subpopulation and the others, with the highest value ($G_{st}$ = 0.372) between the XJ and GW subpopulations, followed by the XJ and YC subpopulations ($G_{st}$ = 0.352), which was the lowest compared to the SNW subpopulation ($G_{st}$ = 0.248). Among the southwestern subpopulations, genetic differentiation was almost negligible; the maximum was between the XZ and YSE subpopulations ($G_{st}$ = 0.139), and the minimum was between the YC and YNE subpopulations ($G_{st}$ = 0.012). All $P$ values were significant ($P$ < 0.01). In groups E and L (Fig. 5C and D; see also Table S3C and D), greater genetic differentiation only existed among subpopulations with elevations above 2,000 m, and $G_{st}$ ≥ 0.15 mainly occurred in pairwise comparisons among the subpopulations 1.0-1.2K, 2.4-2.6K, and 2.8-3.0K (Fig. 5C). The highest value ($G_{st}$ = 0.254) was between the subpopulations 2.8-3.0K and 1.0-1.2K, and the lowest values were between the subpopulations 1.2-1.4K and 0.2-0.4K and 0.4K-0.6K ($G_{st}$ = 0.012). Compared to group E, group L had lower genetic differentiations among subpopulations; the highest $G_{st}$ value was the

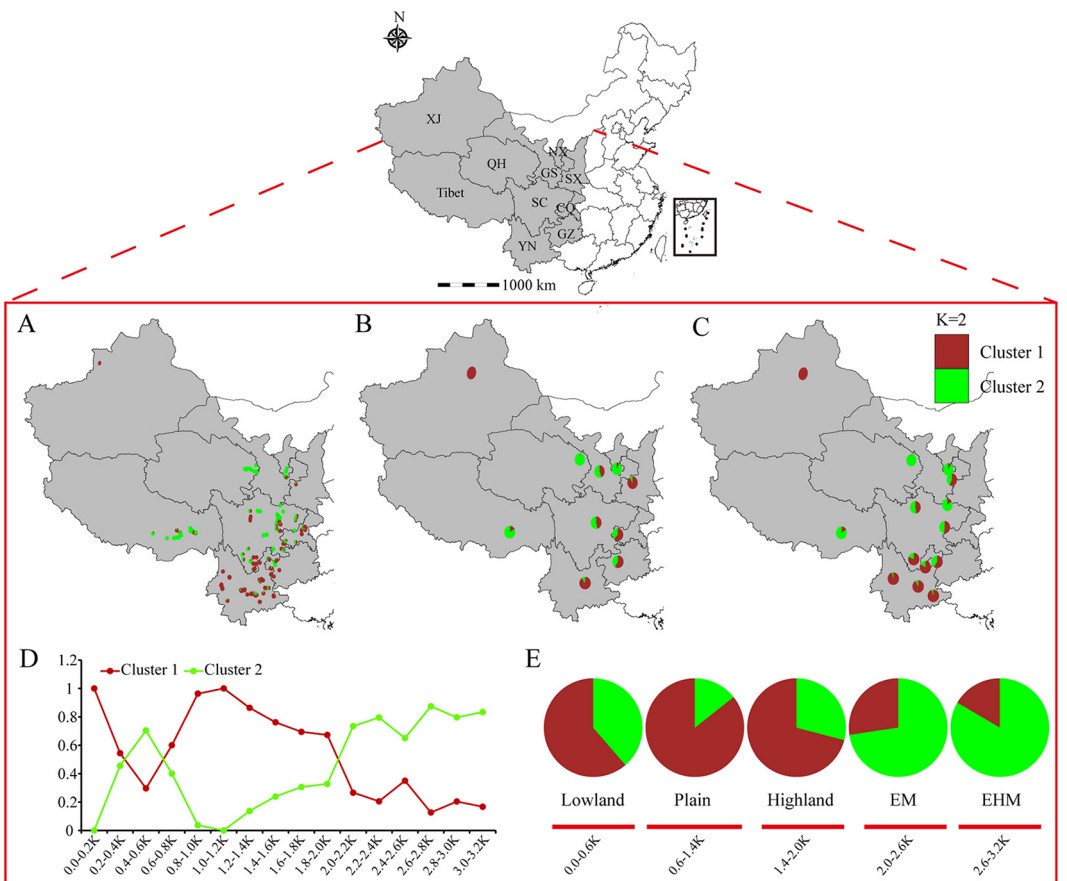

**FIG 3** Geographical distributions of MGs on the horizontal (groups P and R) and vertical (groups E and L) dimensions generated from the nonparametric model discrimination performed in DAPC. When K = 2, the geographical distributions of MG memberships display the proportions of individuals assigned to the brown and green MGs at each sampling site (A), province (B), region (C), elevation (D), and landform (E).

HM subpopulation (compared to the plain subpopulation) ($G_{st}$ = 0.105), and greater genetic differentiations were detected only when the plain subpopulation was compared to the HM and EHM subpopulations (Fig. 5D). However, all $P$ values suggested significant differentiation ($P < 0.01$).

Analysis of molecular variation (AMOVA) demonstrated the highest variation among isolates with 73.82, 74.18, 74.36, 72.91, and 83.74% (Table 9) in groups C, P, R, E, and L, respectively, followed by subpopulations with corresponding values of 19.90, 19.29, 18.36, 20.60, and 11.67%, respectively, and lowest within subpopulations with corresponding values of 6.29, 6.53, 7.29, 6.50, and 4.59%. Except for the insignificant $P$ values among subpopulations in groups C, P, R, and E, other $P$ values indicated significant variations ($P < 0.01$).

The results depicted the genetic separation in the subpopulations from the SW and NW regions from broad to fine spatial scales in both horizontal and vertical orientations, suggesting greater genetic variation between subpopulations. However, the isolates from the SC subpopulation, particularly northern and central, in addition to the central YN and Tibetan subpopulations, were mixed with other subpopulations, indicating the function of those subpopulations in gene interchanges such as bridges connecting the SW and NW regions.

**Reproduction mode in the spatial groups.** In group C, the rbarD values ranged from −0.125 in the SC-ML subpopulation to 0.596 in the SC-YY subpopulation (Fig. 6; see also Table S4A), and the random distribution of rbarD revealed 17 (31.48%) county-wide subpopulations with $P \geq 0.05$, indicating linkage equilibrium and most likely sexual reproduction. In group P, the rbarD values ranged from 0.019 in the SC subpopulation to 0.209 in the SX subpopulation (Fig. 6; see also Table S4B), and two (20%) provincial subpopulations (SX and GZ) had $P \geq 0.05$. In group R, the rbarD values ranged from 0.023 in the GW

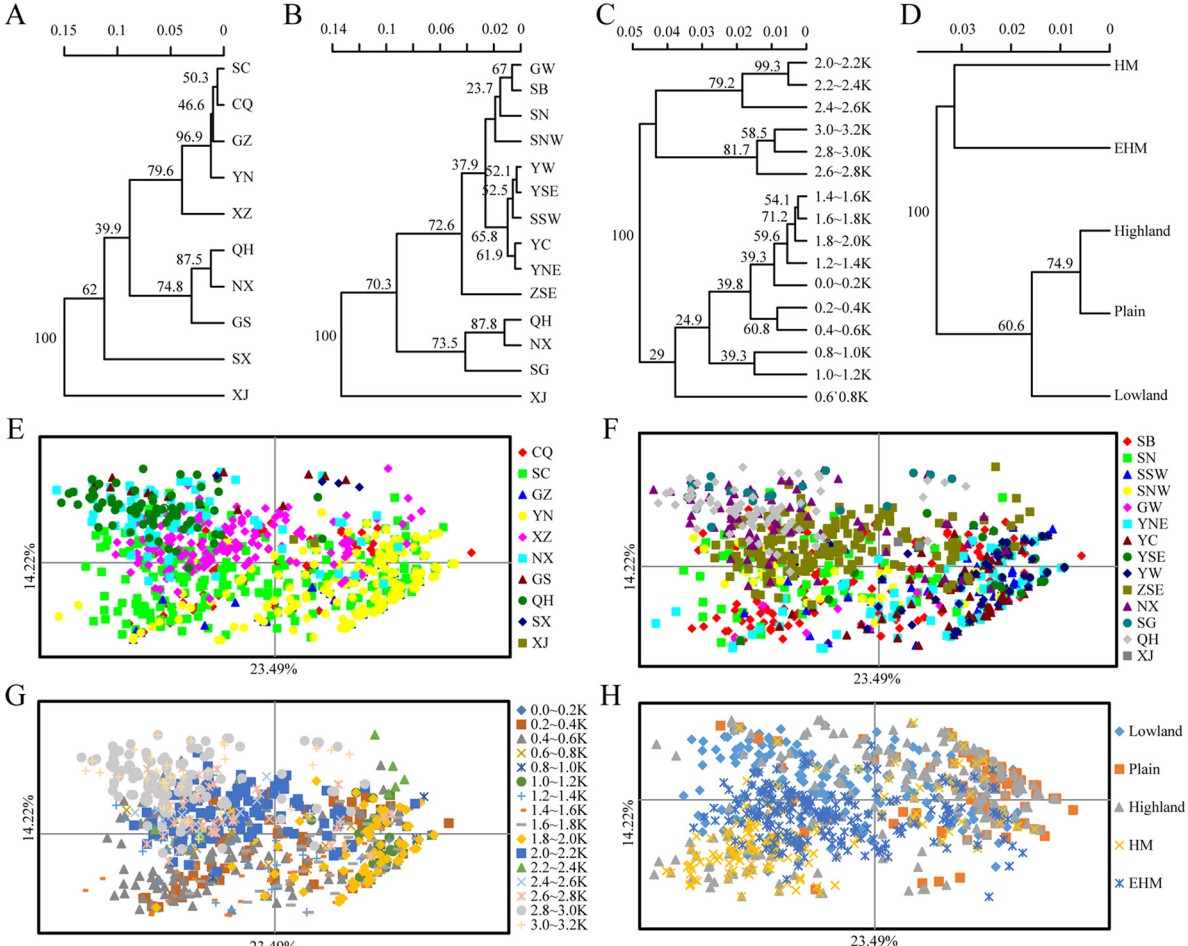

**FIG 4** Clustering of neighbor-joining (NJ) trees and PCoA. (A) The provincial NJ tree shows four clusters based on a distance of 0.05. (B) The regional NJ tree identified three clusters based on a distance of 0.005. (C) The elevational NJ tree confirmed three clusters when the distance was 0.04. (D) The geomorphic NJ tree displayed three clusters when the distance was 0.02. (E) PCoA of 2,103 *Pst* isolates at the provincial level. (F) PCoA result for group R. (G) PCoA result for group E. (H) PCoA result for group L.

subpopulation to 0.105 in the SG subpopulation (Fig. 6; see also Table S4B), and only the GW subpopulation had $P \geq 0.05$ for sexual reproduction. In group E, the rbarD values ranged from 0.013 (1.0-1.2K) to 0.264 (0.8-1.0K) (Fig. 6; see also Table S4B), and $P \geq 0.05$ was detected only in 1.0-1.2K. No geomorphic subpopulation was detected for sexual reproduction (see Table S4B). These results revealed the possibilities for sexual reproduction of *Pst* populations in the SW and NW regions but primarily asexual reproduction. It is important to note that the switch from the broad spatial scale to the fine spatial scale led to differences in the detection of reproduction modes; the highest sexual reproduction signals were detected in the county-wide subpopulation, followed by the provincial subpopulations and only one regional and elevational subpopulation. However, there was no sexual reproduction detected in group L.

**Correlation between geographical distances and genetic distances.** The Mantel test of the five spatial groups (groups C, P, R, E, and L) showed a significant positive correlation between geographic distances and genetic distances. The corresponding values of *R* (correction coefficient) and *P* (significance) were 0.149 and 0.01, 0.124 and 0.01, 0.141 and 0.01, 0.137 and 0.01, and 0.120 and 0.001, respectively, indicating that the genetic variation increased as the geographic distance increased. However, the five low *R* values indicated low genetic variation among the five spatial subpopulations.

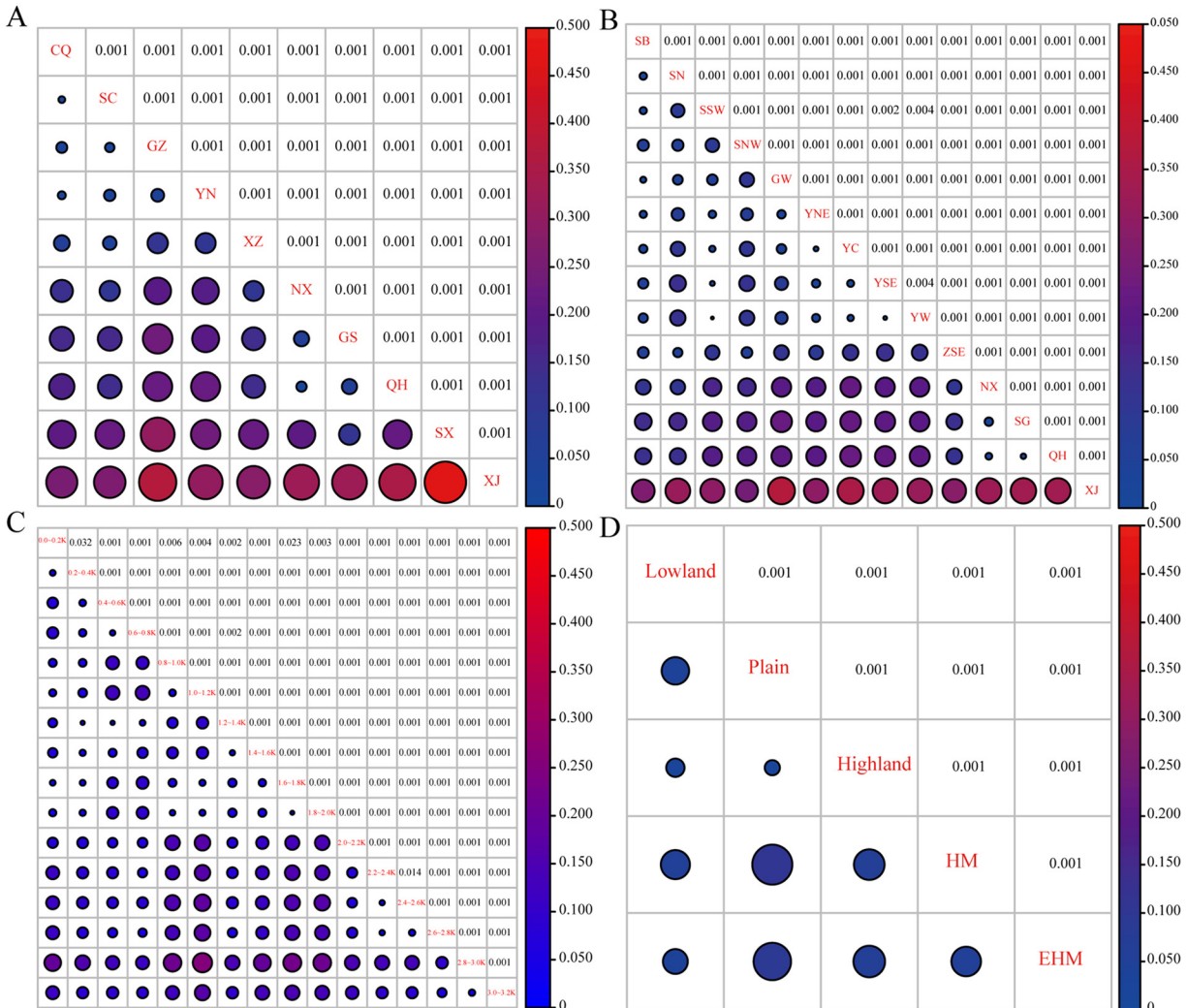

**FIG 5** Population differentiation of 2,103 *Pst* isolates in the four spatial groups indicated by $G_{st}$ values based on G-Statistics analysis. The shade of color and the size of the circle represent the values. (A) $G_{st}$ values for group P. (B) $G_{st}$ values for group R. (C) $G_{st}$ values for group E. (D) $G_{st}$ values for group L.

## DISCUSSION

Making clear the genetic relationships and population dynamics of species among different geographic areas on a large scale is crucial to elucidate the effects of epidemiology on plant disease (32, 33). *Pst* is a parasitic fungus with highly variable geographic subpopulations, thus leading to the need to increase monitoring of its population genetics (4, 6–8, 11–13, 31). Our study elucidated the genetic diversity and connectivity, the spatial structure and geographical distribution, the reproduction mode, and the correlation between genetic distance and geographic distance. These findings provide a clear genetic relationship of the *Pst* population in key epidemic regions in China and will help to formulate adequate management strategies for wheat stripe rust disease and boost the robustness and reliability in understanding the evolutionary mechanisms in *Pst*.

**Abundance of isolates as a foundation for phylogeography.** Population genetics is a robust method to explain the evolution occurring within a certain species (9, 34–37). Sampling strategies can affect a study (38), and extensive sampling over large areas could be a precondition for any phylogeographic research (38, 39). The object of this study was to analyze the migration and colonization properties of the *Pst* pathogen, since it has been demonstrated that urediniospores can disperse thousands of miles across the region by upper air currents (2, 4–7, 40). In previous studies, relatively smaller numbers of *Pst* isolates (the numbers of

**TABLE 9** AMOVA for the five spatial groups based on the corresponding clone-corrected data sets

| Spatial groups | Source of variation | Df | Sum of squares | Mean squares | Estimated variance | Variation (%) | $P^a$ |
|---|---|---|---|---|---|---|---|
| Group C | Among counties | 14 | 694.309 | 49.593 | 0.522 | 19.896 | 0.145 |
| | Within counties | 1 | 17.295 | 17.295 | 0.165 | 6.287 | 0.001 |
| | Among isolates | 1,086 | 2,102.037 | 1.935 | 1.936 | 73.818 | 0.001 |
| | Total | 1,101 | 2,813.641 | | 2.622 | 100 | |
| Group P | Among provinces | 14 | 575.702 | 41.122 | 0.531 | 19.292 | 0.136 |
| | Within provinces | 1 | 16.066 | 16.066 | 0.179 | 6.528 | 0.001 |
| | Among isolates | 864 | 1,765.333 | 2.043 | 2.043 | 74.180 | 0.001 |
| | Total | 879 | 2,357.101 | | 2.753 | 100 | |
| Group R | Among regions | 14 | 613.350 | 43.811 | 0.493 | 18.356 | 0.145 |
| | Within regions | 1 | 17.243 | 17.243 | 0.196 | 7.287 | 0.001 |
| | Among isolates | 951 | 1,897.942 | 1.996 | 1.996 | 74.357 | 0.001 |
| | Total | 966 | 2,528.535 | | 2.685 | 100 | |
| Group E | Among elevations | 14 | 689.520 | 49.251 | 0.552 | 20.596 | 0.133 |
| | Within elevations | 1 | 16.989 | 16.989 | 0.174 | 6.496 | 0.001 |
| | Among isolates | 1,019 | 1,991.021 | 1.954 | 1.954 | 72.907 | 0.001 |
| | Total | 1,034 | 2,697.530 | | 2.680 | 100 | |
| Group L | Among landform regions | 4 | 276.022 | 69.006 | 0.322 | 11.669 | 0.001 |
| | Within landform regions | 11 | 88.019 | 8.002 | 0.127 | 4.589 | 0.001 |
| | Among isolates | 884 | 2,045.831 | 2.314 | 2.314 | 83.742 | 0.001 |
| | Total | 899 | 2,409.873 | 2.681 | 2.764 | 100 | |

$^a$Significance tests were performed using 1,000 permutations.

samples collected was ranging from 94 to 961) collected from a few provinces were characterized (17–19, 24–31), but there is no large spatial scale genetic study of *Pst* isolates from continuous wheat-growing areas. Therefore, to analyze the genetic relationships within the *Pst* population in the key epidemic regions of China, sampling over large areas was a prerequisite. In our study, we collected 2,103 *Pst* isolates from vast wheat production areas in the two major wheat production agroecosystems in SW and NW China. Since these areas are two key epidemic regions with different topographies, extensive sampling in these regions should produce reliable and meaningful results for the characterization of the *Pst* population in China.

Sampling approaches may be different due to different study purposes and hierarchical analyses (38). In the present study, we divided the full range of the spatial scale into horizontal and vertical dimensions: the broader horizontal spatial scale included 14 regions and 10 provinces, and the fine horizontal spatial scale included 63 sampling counties and 261 sampling sites. Likewise, the vertical spatial scales included 16 elevational and 5 geomorphic subpopulations. The recommended sample sizes for a subpopulation and population are approximately 10 to 20 individuals and 25 to 30 individuals, respectively (38, 41). Except for one sampling site in the Shaanxi and Xinjiang Provinces, all other samples were representative and evenly distributed at each spatial level. The reason for the lower number of samples in the two aforementioned provinces was that the local wheat planting area decreased year by year, and due to implemented chemical control, we missed the best sampling period.

Nevertheless, based on the genetic differentiation and genetic structure at the provincial level, the revealed relationships between the two provincial subpopulations and other subpopulations were consistent with the results of previous studies (i.e., the genetic relationships were closest between the Shaanxi and Gansu subpopulations (27), and the Xinjiang population was more independent and closer to the northwestern population (29), which was only affected in the case of the provincial genetic diversity analysis). Therefore, the results are still valuable. Hence, a good sampling strategy is vital for phylogeographic analysis of population genetic relationships.

**Genetic homogeneity of *Pst* in SW and NW China.** The geographical homogeneity and origin of species are always hot topics and difficult targets in the field of population genetics. Genetic diversity and genotypic composition could be used to assess genetic homogeneity. Flight-capable organisms forced by air currents or ocean

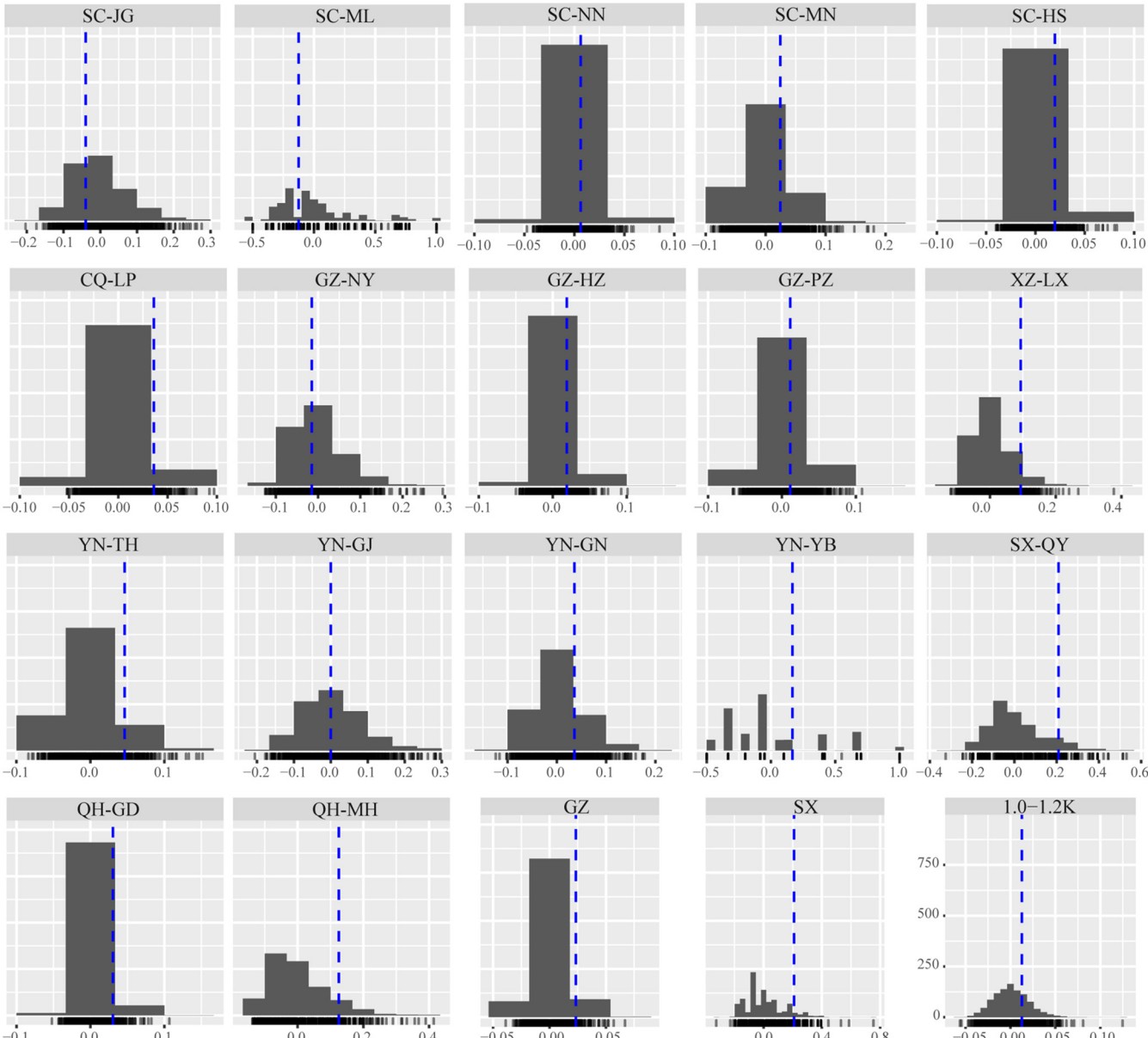

**FIG 6** Histograms showing the subpopulations with the possibility of sexual recombination. The rbarD values calculated from 1,000 permutations based on the corresponding clone corrected data sets. The observed rbarD values are shown as vertical dashed lines inside the distributions showing the absence of linkage disequilibrium, indicating sexual reproduction.

currents are more widely distributed (42–44). Previous studies ignored the continuity in the natural distribution of hosts for *Pst* and the mobile characteristics of the airborne fungus. Hence, it is vital to interpret their evolution and analyze their genetic diversity throughout the various ecological areas they are found in. Ali et al. confirmed that the genetic diversity was highest in the Himalayas and their neighboring regions and then proposed that this area could be the origin of *Pst* that is now found worldwide (9). Coincidentally, we found that regional *Pst* subpopulations with relatively higher genetic diversities appeared to occur in the Qinling Mountains, as the east slope was near the Himalayan Mountain range, as well as the junction areas of the Qinghai-Tibet Plateau, such as Qinghai, Ningxia, Gansu, southeastern Tibet, and northern and central Sichuan. The lowest diversity was found in the southernmost Yunnan and Guizhou Provinces, and these findings were consistent with those of many previous studies (17–19, 24–31). Genotypic composition could reflect genetic diversity; the population that contained

more private MLGs indicated higher genotypic diversity; likewise, the more shared MLGs, the lower genotypic diversity. We found that the dominant and shared MLGs emerged in southwestern China; in contrast, the northwestern subpopulation contained few dominant any MLGs and were mainly composed of private MLGs. Currently, the higher genetic diversity in northwestern China is widely accepted.

Northwestern Tibet, northern Sichuan, and the Sichuan Basin showed higher genetic diversity, possibly because these regions are adjacent to each other and have similar geographical and climatic characteristics. In particular, the high-altitude, mountainous areas near the Qinghai-Tibet Plateau and the Qinling Mountains are important oversummering regions that are the sources of new races (4, 6, 8, 11–14). *Pst* individuals with high variation that survive and encounter suitable environmental conditions for infection can expand their populations rapidly (2, 4, 6, 8, 11–14, 16, 40). The number of private genotypes increases with increased abundance and leads to more individuals with higher variations of genetic diversity; the vertical subpopulations possessed the highest genetic diversity in our study, confirming that the subpopulations that include the oversummering region show high levels of genetic diversity (4, 6, 8, 11, 15, 17–19, 24–31). Another possible reason for the higher genetic diversity in the northwestern region was the planting of diversified resistant cultivars over large areas. In the northwestern region, especially in Gansu Province, the local agricultural departments have configured wheat cultivars with as many resistances as possible to control the disease. Therefore, the diversity of background resistance readjusted the pathogenic population structure (45).

The lower genetic diversity in the southwestern subpopulations of southwestern Sichuan, northeastern Yunnan, and western Guizhou was probably due to similar geographic environments, climate conditions and cropping systems. The warm and rainy spring seasons in these areas are suitable for rust migration and colonization, thus generating many repeat genotypes and reducing the overall genetic diversity. Therefore, we observed more dominant and shared genotypes in these regions.

Interestingly, the relatively higher genetic diversity in Tibet shown in the present study was inconsistent with the low genetic diversities reported by Ma et al. and Hu et al. (30, 31). This difference could be caused by differences in the numbers of samples, sampling years, sampling locations, and loci. The same differences in genetic diversity have been reported in other regions. The *Pst* population in Gansu had much higher diversity than those in Xinjiang and Qinghai (29). Conversely, Zhan reported that the population in Xinjiang had a higher genetic diversity than the populations in Qinghai, Gansu, Ningxia, and Tibet (24). The other possibility for the difference in genetic diversity in the same region was driven by the introduction and colonization of potential migrants. This hypothesis failed to be tested in the present study due to 1-year collections. Characterization of annual *Pst* collections from these regions and genotyping with the same set of molecular markers should be conducted to identify the newly introduced genotypes and to monitor the dynamics of *Pst* populations over the years.

**Genetic heterogeneity of *Pst* in SW and NW China.** Genetic variation is believed to increase the level of heterogeneity (46–48). The mechanisms for generating genetic variation in *Pst* include mutation, somatic recombination, and sexual reproduction (4, 49, 50). The maximum source of variation was within *Pst* isolates, consistent with previous studies (10, 17, 25, 27, 28, 51), and supported by the present study that showed 72% of the total variation within individuals. Thus, these results suggest the possibility to analyze differentiations in the genetic diversity of *Pst* from different geographic areas.

The population spatial structures generated from DAPC and STRUCTURE analyses, neighbor-joining trees, and G-statistics explained, to a certain extent, the subdivision of *Pst* populations between the SW and NW regions. DAPC and STRUCTURE clustering revealed the variation in MGs originating from the northwest and southwest regions. The significant positive correlation between genetic distances and geographic distances supported the above findings. The provincial PCoA result and neighbor-joining tree revealed a close genetic relationship among *Pst* populations within southwestern and

northwestern Guizhou, especially Chongqing and Sichuan, Guizhou and Yunnan, and Qinghai and Ningxia (18, 19, 26, 52), as well as observed in the regional subpopulations of western Guizhou close to northern Sichuan, northwestern Sichuan, and the Sichuan Basin, while the southwestern Sichuan population grouped with the Yunnan population. The above genetic relationships were supported by the four molecular groups implemented in GENELAND based on 300 random clone-corrected genotypes (see Fig. S3). The results calculated by G-statistics confirmed that genetic differentiation greater than 0.25 only existed in subpopulations between the southwestern and northwestern regions and between the Xinjiang population and all inland populations. However, there was almost no genetic differentiation within the southwestern and northwestern regions, suggesting different levels of gene exchange within different subpopulations. The low coefficient of genetic differentiation between the Tibetan, northern and northwestern Sichuan, and Sichuan Basin populations suggested that there were inoculum exchanges. Both northwest Sichuan and Tibet are extremely high-altitude mountainous areas, and Tibet also contains abundant oversummering regions (30, 31). There was probably inoculum exchange at this junction. Similarly, the negligible genetic differentiation between the southwestern Sichuan and Yunnan populations suggested frequent inoculum exchange within these regions.

These consequences are probably due to air currents and geographic conditions. In spring, prevailing southern winds occur in southwestern China (20). Li et al. observed the trajectory of upper airflow in Yunnan during the peak period of wheat stripe rust and found that in Yunnan, it could cross the Sichuan Basin and even reach the Longnan region of Gansu (20). The inocula carried by the upper air currents spread to Guizhou, Sichuan, Chongqing, and other adjacent areas, and then these inocula fused with each other, which was one of the reasons for the low genetic differentiation in the southwestern region. The genetic differences between the northwestern and Yunnan-Guizhou populations were greater than that of the Sichuan-Chongqing population, which was speculated to be caused by the gene interchange between the populations in the border areas of northern Sichuan, southern Gansu, and northern Shaanxi (25–27).

It was reported that the Xinjiang region is far inland (30, 31), and the present result still supports this view. Although there were few isolates from the Xinjiang region in this study, the findings could provide some clues to explore the genetic relationship between the Xinjiang population and the inland population. There may be two potential possibilities to explain the furthest genetic distance between the Xinjiang population and the inland population. First, the host wheat closest inland is distributed in Qinghai and in extremely high-altitude, mountainous areas of Tibet, and the discontinuity in host distribution reduced the gene exchange between the *Pst* populations of Xinjiang and inland inocula. Second, it was recorded that the Xinjiang population was affected by the Ili Valley, and researchers speculated that there were gene interchanges with foreign *Pst* populations (30, 31). Long-term reproductive isolation would lead to consolidation of the spatial genetic structure, which reflects the genetic relationships between different populations (23).

**Mixture reproduction modes of *Pst* in SW and NW China.** In recent decades, numerous studies have focused on the sexual reproduction of *Pst*, especially in the oversummering region. Sexual reproduction plays a certain role in epidemics, but its specific function remains to be further explored (53). *Berberis* spp. has been found in Gansu, Qinghai, Ningxia, Shaanxi, Sichuan, Yunnan, Tibet, and Guizhou Provinces (4, 18, 26, 29, 53, 54), and our study revealed linkage disequilibrium sexual reproduction in many county-wide subpopulations in all provinces, indicating the possibility of sexual reproduction. Nevertheless, no sexual reproduction presented in group L was caused by a massive number of individuals in one subpopulation, and the redundant genotypes affected linkage disequilibrium detection (55). These findings provide clues for the follow-up investigation of the distribution range of alternate hosts and thus provide theoretical guidance for implementing effective management of wheat stripe rust.

In conclusion, in this study, we collected 2,103 *Pst* isolates from vast wheat planting areas that are considered to have a high occurrence of wheat stripe rust disease in China, documented the genetic relationships among the *Pst* subpopulations from different geographic

areas under broad to fine spatial scales, confirmed the highest genetic diversity in the northwestern region and lowest in Yunan and Guizhou, and demonstrated different extents of genetic variation on horizontal and vertical scales that ranged from south to north and north to south, low to high altitude, or vice versa. Moreover, three large provincial spatial structures of southwestern, northwestern, and Xinjiang were revealed, and considerable genetic variation was observed between the SW and NW regions, Xinjiang, and other regions. All the findings displaying the comprehensive and clarified genetic relationships on a large scale offer molecular evidence for guiding wheat stripe rust prediction and management. Nevertheless, *Pst* collections from multiple years are needed for further studies to identify consistent population genetic dynamics and evolutionary mechanisms of the pathogen.

## MATERIALS AND METHODS

**Sampling and grouping.** From January to June 2019, a total of 2,103 *Pst* isolates were collected from SW and NW China (Fig. 1; see also Table S2). The southwestern provinces or cities consisted of Yunnan (YN), Sichuan (SC), Chongqing (CQ), Guizhou (GZ), and Tibet (XZ or Tibet), and the northwestern regions consisted of Gansu (GS), Ningxia (NX), Qinghai (QH), Shaanxi (SX), and Xinjiang (XJ). The sampling strategy deployed in different provinces was based on different planting areas and disease occurrence, and in general, the samples from one county were used as the minimum subpopulation, selecting approximately 10 towns that were 5 to 20 km apart from each other in one county, selecting five to 10 villages that were 3 to 10 km apart from each other in one town, selecting five fields that covered all planting areas as much as possible in one village, selecting five distant centers of WSR in one field and picking three leaves with single-stripe uredinia from the edge of each center. Information such as the latitude and longitude of the sampling site, sampling date, cultivar, elevation, growth period, stripe rust severity, wheat growth stage, disease occurrence date (see Fig. S1A to D), severity and incidence of the disease, and overwinter and/or oversummer status was recorded.

Based on the background of the samples, we assigned all isolates into three horizontal subpopulations and two vertical subpopulations from broad to fine spatial scales (see Table S2). Horizontal spatial scales were grouped by county (group C), with a total of 63 countywide subpopulations; by province (group P), with 10 subpopulations; and by region with geographical proximity and topography (group R), with 14 subpopulations. Vertical spatial scales were grouped by elevation (group E), an elevation of 200 m being used as the criterion for population classification in group E, with 16 subpopulations ranging from 0.0-0.2K to 3.0-3.2K, and by landform (group L); an elevation within 0 to 600 m was considered lowland, 601 to 1,400 m was considered plain, 1,401 to 2,000 m was considered highland, 2,001 to 2,600 m was considered HM (high mountain), and 2,601 to 3,200 m was considered EHM (extremely high mountain).

**Production and preservation of *Pst*.** The wheat leaves were kept in a desiccator at 4°C for 2 days. After drying, they were enfolded within absorbent paper. Some of the leaf samples with fresh urediniospores were used to propagate spores, and purification was carried out as previously described (26). Urediniospores were transferred into a 0.5-mL Eppendorf tube and then stored at −20°C until future use.

**DNA extraction and SSR amplification.** Genomic DNA of *Pst* from urediniospores was extracted with cetyltrimethylammonium bromide (CTAB) (29) from infected leaf tissues (~2 cm) using the method of Ali et al. (56). A NanoDrop 2000 spectrophotometer (Thermo Fisher Scientific, Waltham, MA) was used to measure the DNA concentration and determine the quality based on the absorptions at wavelengths of 260 and 280 nm. DNA samples were diluted to 50 ng/$\mu$L for SSR amplification. Twelve pairs of published SSR primers (57–60) were used to genotype every isolate (see Table S1). The materials and specific experimental procedures described by Wang et al. (51) were followed.

The alleles were manually scored for each pair of SSR markers based on the size of the PCR products using GeneMarker V2.2 (Softgenetics, State College, PA), and the isolates with identical alleles across all 12 pairs of SSR markers were assigned to the same multilocus genotypes (MLGs). Samples with any missing alleles were eliminated from the 2,702 samples, and 2,103 samples remained.

**Population genetic diversity analysis.** To assess the number of selected loci that were sufficient for subsequent population genetic analysis, a genotype accumulation curve was employed. The genotype curve reached 100%, indicating that the number of loci selected was sufficient for the following analysis (55).

The genetic diversity of each locus was evaluated by $N_a$ (the observed alleles), $H_o$ (the observed heterozygosity), and $H_e$ (the expected heterozygosity) calculated using GENALEX 6.501 software (61–63). The allele frequencies were used to calculate the polymorphism information content for each locus with the program PIC-CALC (64). The genotype accumulation curve was created in R program version 4.0.2 with the Poppr package using 1,000 bootstrap replicates to explain the minimum number of loci necessary to discriminate among individuals (55).

Genotypic diversity estimated for the spatial groups (groups P, R, E, and L) was determined by the "Poppr" package in R program version 4.0.2 (55). Following the indices calculated, the number of individuals observed (*N*), the number of MLGs observed, the ratio of MLGs over *N*, and the number of expected MLGs (eMLGs) were determined as an approximation of the number of genotypes, which would be expected at the largest shared sample size based on rarefaction and the shared sample size (65). The Shannon-Wiener index (*H*), which measures MLG diversity (66), was calculated as the proportion of different genes among populations (67). Stoddart and Taylor's index (*G*) indicates genotypic richness (68). Lambda Simpson's index ($\lambda$), which indicates MLG richness and uniformity, and the corrected Simpson's index (corrected $\lambda$) were calculated to assess richness, avoiding the influence of different population sizes (69). Genotypic evenness (E.5) indicates the distribution of alleles at each locus (70–72). Gene diversity estimated for the spatial groups (groups P, R, E, and L) was determined by $H_o$, $H_e$, $N_a$, and $N_e$ (the effective

alleles); private alleles ($A_p$), which represent the alleles unique to a single population; and polymorphic loci (PL), which indicate the diversity of loci. All the indices were conducted in the program GENALEX 6.501 (62, 63).

The level of genetic homogeneity was determined by the structure of genotype compositions, and the 10 most repeated MLGs were defined as the dominant MLGs. Considering that the MLGs that occurred at least twice as frequently as the MLGs shared within or among subpopulations, the remaining MLGs were private MLGs that were unique to the subpopulation. We calculated the proportions of dominant, shared, and private MLGs in each spatial group in Excel and generated the corresponding histogram, as well as the number and frequency of the shared MLGs within and among subpopulations.

**Population subdivision on various spatial scales.** Both no-model-based and Bayesian model-based hierarchical cluster methods were used to identify putative genetic clusters to estimate the population structure in various spatial groups. The no-model-based method was conducted in the R program 4.0.2 based on dissimilarities by the ward.D2 method of the "hclust" function (73), combined with the multivariate discriminant analysis of principal components (DAPC) by the DAPC function in the Adegenet package (74, 75). All isolates were assigned to the molecular groups (MGs) according to the cluster (K), and a scatterplot was generated to exhibit the membership probability (76). We calculated the proportions of the different MGs in the different spatial subpopulations and generated corresponding pie charts that were then displayed on the map to explain the geographical distribution of population genetic structure. The other Bayesian model-based cluster analysis was implemented in STRUCTURE 2.3.4 (77) to identify clusters and assess the extent of admixtures. The putative K clusters ranging from 1 to 10 were determined by repeating 15 independent runs, using MCMC replications with 10,000 iterations, and a burn-in period of 2,000. The resulting files were uploaded to STRUCTURE HARVESTER (http://taylor0.biology.ucla.edu/structureHarvester) and evaluated for the ΔK and the maximum value as the best K (78). Then, CLUMPP 1.1.2 and Distruct 1.1 were used to generate a bar plot to demonstrate the clusters (77, 79, 80).

To gain insights into genetic differentiation and the spatial population structure, multiple approaches were used. First, based on Nei's genetic distance of groups P, R, E, and L, a bootstrap analysis was performed using the function "aboot" in the Poppr package with 1,000 replicates to generate the corresponding neighbor-joining trees. Second, nonparametric principal coordinate analysis (PCoA) was conducted by GENALEX 6.501 to visualize the genotypic interchange among all genotypes detected in the various spatial groups (62, 63, 81). The third measurement used to assess population differentiation was pairwise G-Statistics analysis ($G_{st}$) in GENALEX 6.501 software to generate a $G_{st}$ value matrix that was visualized as a heatmap by using the ggplot package in the R program (62, 82). After deletion of the same genotypes within spatial groups, the remaining numbers of isolates in groups C, P, R, E, and L were 1,102, 880, 967, 1,035, and 900, respectively. The last method used was molecular variance analysis (AMOVA) in the Poppr package by using the function "Poppr.amova" to partition genetic variance with 1,000 permutations, in which the amounts of each variance source based on the corresponding clone corrected data sets were calculated (55, 83).

**Reproduction mode detection.** Linkage disequilibrium was tested by the standard index of association (using rbarD to force the index of association [$I_A$] to lie between 0 [linkage equilibrium] and 1 [full disequilibrium]) calculated in the Poppr package of R program 4.0.2 based on the clone corrected data sets to indicate the reproduction mode (84). The null hypothesis rbarD = 0 was tested using 1,000 permutations to indicate no linkage among loci, and the observed rbarD was distributed in the randomized data set, and the $P$ value was nonsignificant ($P \geq 0.05$). Meanwhile, the population recombined freely with sexual reproduction (85). An rbarD of 1 indicated significant linkage disequilibrium ($P < 0.05$) among loci and reproduced genotype by cloning, while the observed rbarD value was located outside the distribution of the randomized data set (55).

**Association between genetic distance and geographic distance.** A Mantel test was conducted in GENALEX 6.501 with 1,000 permutations based on the corresponding clone-corrected data sets to assess the significance of the association between genetic distance and geographic distance. A pairwise, individual-by-individual genetic distance matrix was generated from binary data (86), and the geographical distance was calculated from longitude and latitude.

**Data availability.** The frequencies of all *Pst* genotypes generated by 12 SSR loci are documented in Table S5 in the supplemental material.

## SUPPLEMENTAL MATERIAL

Supplemental material is available online only.
**SUPPLEMENTAL FILE 1**, PDF file, 6.6 MB.
**SUPPLEMENTAL FILE 2**, XLSX file, 0.5 MB.

## ACKNOWLEDGMENTS

This study was supported by the National Key Research and Development Program of China under grant 2021YFD1401000.

We have no conflicts of interest to disclose.

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
