## [Reviewer comments · Microbiology Spectrum]

Microbiology Spectrum

The genetic relationship of *Puccinia striiformis* f. sp. *tritici* in southwestern and northwestern China

Bingbing Jiang, Cuicui Wang, Cunwu Guo, Xuan Lv, Wenfeng Gong, Jie Chang, Hongpan He, Jing Feng, Xianming Chen, and Zhanhong Ma

Corresponding Author(s): Bingbing Jiang, China Agricultural University

Review Timeline:

Submission Date:	April 27, 2022
Editorial Decision:	May 23, 2022
Revision Received:	July 1, 2022
Accepted:	July 8, 2022

Editor: Giuseppe Ianiri

Reviewer(s): Disclosure of reviewer identity is with reference to reviewer comments included in decision letter(s). The following individuals involved in review of your submission have agreed to reveal their identity: Xupo Ding (Reviewer #1); Yuxiang Li (Reviewer #2); Mingju Li (Reviewer #3)

Transaction Report:

DOI: <https://doi.org/10.1128/spectrum.01530-22>

May 23, 2022

Prof. Zhanhong Ma
China Agricultural University
Beijing
China

Re: Spectrum01530-22 (The genetic relationship of *Puccinia striiformis* f. sp. *tritici* in southwestern and northwestern China)

Dear Prof. Zhanhong Ma:

Link Not Available

Sincerely,

Giuseppe Ianiri

Journals Department
Reviewer comments:

Reviewer #1 (Comments for the Author):

Dear editors,

The manuscript entitled with the genetic relationship of *Puccinia striiformis* f. sp. *tritici* in southwestern and northwestern China have been yet reviewed. This manuscript is an extremely rare research about large population diversity analysis of Pst from oversummering and overwinter areas in recently. Obviously, this manuscript presented the spatial genetic relationship, population structure and phylogeography with various geographical Pst sub-population genetics via SSR with massive data from extensive regions of different altitude and will contribute to predict and manage wheat stripe rust in many regions of wheat cultures, including China. The data was comprehensively analyzed and population subdivision analysis was particularly excellent. The methods and technologies used in this manuscript are precise, scientific and various, but some details should be improved or appended in the results and analysis. The comments present in the following could be referenced by editors.

Please noticed that the comments present in the following were based on the document named 278085_0_278085_0_merged_1651065866.pdf from review system.

1. Lots of abbreviation were employed in the Figs, so each abbreviation should be list in each figure information or add the summary including all abbreviations at the front or end of manuscript.
2. For consistency, the legend presented as 1,2,3 might be replaced as cluster 1, cluster 2, cluster3 in fig2.
3. The fig4B-4C, the values in the branch should be improved via reducing the font size.
4. The values showed at bottom left in Fig5 should be shifted to the bottom of each roundness and it can be realized via Adobe Illustrator.
5. The table1 should be transfer in the additional files.
6. The original title is general or ambiguous, it should be revealed the major results or conclusion to increase attractiveness. Such as, the genetic diversity characterization of *Puccinia striiformis* f. sp. *tritici* in southwestern and northwestern China with 2103 isolates or The genetic diversity of *Pst* in *Puccinia striiformis* f. sp. *tritici* provide new insights into the population dynamics and evolutionary mechanisms of *Pst* in southwestern and northwestern China. These two titles are just suggestion and the final title should be confirmed by the authors and editors.
7. Line 23-26, it is suggested to modified as Wheat stripe rust, caused by *Puccinia striiformis* f. sp. *tritici*, is an important disease for wheat worldwide and constantly threaten wheat production in southwestern or northwestern of China where the areas fit for *Pst* oversummering and overwintering in China.
8. Abbreviations were not suggested in the abstract.
9. Line 81-83, is this the previous research about the potential oversummering or overwintering aeras of *Pst* in China? If yes, please add their result or conclusion.
10. Line 104-118, the description should be summarized and rewrite, the results from the references were simple presented by publish years and this is not professional.
11. Line 156-162, the highest and lowest regions and their diversity index should be presented.
12. The total name of EHM was not list when firstly described in the manuscript.
13. Line189-194, line 311-316 and line 327-333 might be included in the discussion.
14. Line 332-333, there was no sexual reproduction in Group L, why? The reason could be discussed in the discussion.
15. The figure7 was suggested to listed in the supplemental flies.
16. As the focused on the oversummering and overwintering areas, could you furtherly inference or discuss the epidemic pathway of *Pst* in these areas or Chinese regions? This might be contributed to study population dynamics and evolutionary mechanisms of *Pst*.
17. Line 543, Liang (2013) should be removed.
18. If the program or reaction system were same with your previous study, please curtrail line 554-569 as citing a reference.
19. The software and statistics method used in Fig S1 should be revised in the materials and methods.
20. In table S1, the longitude and latitude of sample areas should be better added. This is optional and not necessary.
21. Finally and importantly, all language, include the description of figures, must be improved and modified by naïve academic editor.

Reviewer #2 (Comments for the Author):

Manuscript #Spectrum01530-22 investigated the genetic relationship of *Pst* in southwest and northwest of China. More than 2000 *Pst* samples across major provinces in western China were analyzed from horizontal (i.e., County, provincial and regional subspecies) and vertical (i.e., altitude and geomorphic subspecies). The finding in the study should be useful in dissect the epidemics of stripe rust in China. However, some concerns should be addressed.

Major concerns:

1. The authors classified *Pst* samples into different groups according to different criteria, such as country, province, region, elevation, and landform, which is good for systematical analysis. However, it seems the other groups (E, R, L) are the subgroups of Group P displayed in Table S1. So, it is questionable that classifying the isolates simply by elevation or other single factor is meaningful in the study.
2. In this case, it is hard to understand why the isolates collected from Xinjiang and Shaanxi are excluded in Group P, but are included in Group R, E and L.
3. Elevation is the only factor taken into consideration for Group E and R, so it is better to combine these two groups.
4. In the Line 624, the authors mentioned the K clusters from 1 to 10. In fact, the cluster in the manuscript should be ranged from 2 to 9 as shown in Fig. 2. Also, analyses at higher K (5 to 10) could provide a better interpretation in the complicated population.
5. The connection between text and tables/figure is lacking. For example, QH, SC, NX, QH, CQ, GZ, XZ, and PL in the Line 157 and 159 cannot be found in corresponding tables.
6. It is hard to interpretate result in the genetic differentiation analysis in Group E displayed in Fig. 5C. More information should be provided in the Figure. Based on the results, the authors mentioned "The highest value ($G_{st}=0.254$) was between 2.8-3.0K and 1.0-1.2K". Why the highest G_{st} is not between 0-0.2K to 3.0-3.2K?
7. Grammatical errors and ambiguous descriptions impair the quality of the manuscript, such as Line 50-55, 63-64, 66-67, 92-94, 122-125 in the introduction part. The authors should also check and revise the statements in other parts.

Minor concerns:

1. The author should connect different studies in the line 104 to 118 instead of listing all the results.
2. Provide the accurate P value in the Fig. 7, and keep all the "P" Italic.
3. Improve the quality of Fig. 5 to make every number readable.

Reviewer #3 (Comments for the Author):

Comments

1. Several decades research discovered four over-summering areas in China, namely northwestern, south-western, Xinjiang, and northern over-summering regions. The NW and SW regions are the largest and the most important over-summering areas in China, so, it is vital to study the genetics of Pst in the SW and NW regions to improve the management of WSR in China. However, we can't see any sampling strategy from the manuscript, which is crucial to draw a right conclusion.

2. In Table S1, for Xinjiang and Shaanxi, the samples less than 20, and just from one county, one cultivar. For Ningxia and Gansu, although the samples more than 20, even more than 100, but it from one county. For the four provinces, the representativeness is questionable, can't meet the requirement of statistic. Although the authors remove the Xinjiang and Shaanxi isolates when analyzed the genetic diversity data at the provincial level (Group P), but included as a part in Group R, E, and L. Even though at regional level, samples only from one cultivar and less than 20 samples can't represent a region. Therefore, related results like Xinjiang is of greatest genetic variation compared with other regions, is questionable. Few samples can provide some clues, but to draw a conclusion should be prudential.

3. Too much tables and figures, so need to be simplified. For Table 1 can be removed or list as supplemental table for it has been published before. In Fig. 2 and 3, listed $K=2$, $k=3$, respectively. To my knowledge, just one K value is most appropriate, which one? Select it and remove another. For Fig. 7, as R^2 less than 0.03, very low, no need to make the figure, just mention it in the text. In Fig. 1(A,B,C), 6, and Table 2A,2B,3A,3B, "19S" in all words contained can be removed as all samples collected in 2019 spring, no need to mention always, it looks very prolix.

4. In Fig. 1 C, what's the meaning of the abbreviation of "ZSE" etc. ? In Fig. 1 E, what is that "EHM"? Please note it below the Fig or Table when appeared the first time, do not always repeat thereafter.

5. Line 37-38, from the Abstract, the authors wrote "The regional neighbor-joining tree presented three big spatial structures, SW, NW, and Xinjiang", which has been concluded by previous epidemiological study, this work just confirmed the well-known theory, seems have no any new findings.

6. Line 157, QH appeared twice, should be mistakes. And what is the difference between genotypic diversity and gene diversity?

5. Line 554-566, In Materials and Methods part - SSR amplification, for all 12 markers, the reaction condition is similar only the annealing temperature classified as 56°C and 52°C. The statement seemed too prolix, it can be simplified.

Staff Comments:

Preparing Revision Guidelines

Please return the manuscript within 60 days; if you cannot complete the modification within this time period, please contact me. If you do not wish to modify the manuscript and prefer to submit it to another journal, please notify me of your decision immediately so that the manuscript may be formally withdrawn from consideration by Microbiology Spectrum.

Dear editors,

The manuscript entitled with the genetic relationship of *Puccinia striiformis* f. sp. *tritici* in southwestern and northwestern China have been yet reviewed. This manuscript is an extremely rare research about large population diversity analysis of Pst from oversummering and overwinter areas in recently. Obviously, this manuscript presented the spatial genetic relationship, population structure and phylogeography with various geographical Pst sub-population genetics via SSR with massive data from extensive regions of different altitude and will contribute to predict and manage wheat stripe rust in many regions of wheat cultures, including China. The data was comprehensively analyzed and population subdivision analysis was particularly excellent. The methods and technologies used in this manuscript are precise, scientific and various, but some details should be improved or appended in the results and analysis. The comments present in the following could be referenced by editors.

Please noticed that the comments present in the following were based on the document named 278085_0_278085_0_merged_1651065866.pdf from review system.

1. Lots of abbreviation were employed in the Figs, so each abbreviation should be list in each figure information or add the summary including all abbreviations at the front or end of manuscript.
2. For consistency, the legend presented as 1,2,3 might be replaced as cluster 1, cluster 2, cluster3 in fig2.
3. The fig4B-4C, the values in the branch should be improved via reducing the font size.
4. The values showed at bottom left in Fig5 should be shifted to the bottom of each roundness and it can be realized via Adobe Illustrator.

5. The table1 should be transfer in the additional files.
6. The original title is general or ambiguous, it should be revealed the major results or conclusion to increase attractiveness. Such as, the genetic diversity characterization of *Puccinia striiformis* f. sp. *tritici* in southwestern and northwestern China with 2103 isolates or The genetic diversity of Pst in *Puccinia striiformis* f. sp. *tritici* provide new insights into the population dynamics and evolutionary mechanisms of *Pst* in southwestern and northwestern China. These two titles are just suggestion and the final title should be confirmed by the authors and editors.
7. Line 23-26, it is suggested to modified as Wheat stripe rust, caused by *Puccinia striiformis* f. sp. *tritici*, is an important disease for wheat worldwide and constantly threaten wheat production in southwestern or northwestern of China where the areas fit for *Pst* oversummering and overwintering in China.
8. Abbreviations were not suggested in the abstract.
9. Line 81-83, is this the previous research about the potential oversummering or overwintering aeras of Pst in China? If yes, please add their result or conclusion.
10. Line 104-118, the description should be summarized and rewrite, the results from the references were simple presented by publish years and this is not professional.
11. Line 156-162, the highest and lowest regions and their diversity index should be presented.
12. The total name of EHM was not list when firstly described in the manuscript.
13. Line189-194, line 311-316 and line 327-333 might be included in the discussion.
14. Line 332-333, there was no sexual reproduction in Group L, why? The reason could be discussed in the discussion.

15. The figure7 was suggested to listed in the supplemental flies.
16. As the focused on the oversummering and overwintering areas, could you furtherly inference or discuss the epidemic pathway of Pst in these areas or Chinese regions?
This might be contributed to study population dynamics and evolutionary mechanisms of *Pst*.
17. Line 543, Liang (2013) should be removed.
18. If the program or reaction system were same with your previous study, please curtrail line 554-569 as citing a reference.
19. The software and statistics method used in Fig S1 should be revised in the materials and methods.
20. In table S1, the longitude and latitude of sample areas should be better added. This is optional and not necessary.
21. Finally and importantly, all language, include the description of figures, must be improved and modified by naïve academic editor.

Dear editors,

We appreciate your and other three reviewers' work. We revised the manuscript according to the reviewers' comments one by one, and the details as following:

Reviewer #1

1. Now we list the information of each abbreviation in the Figures and text when first appeared.
2. We changed the legend as cluster 1, 2, and 3 in Fig.2.
3. We reduced the font size in Fig.4B-4C.
4. We deleted the values in the Fig. 5, instead by putting the values in Table S3.
5. We transferred Table 1 into the additional files as Table S1.
6. We think the title should be the one we proposed, because genetic relationship including many aspects, in general, involved genetic diversity, genetic differentiation, and genetic structure. Genetic diversity only one perspective, maybe the two titles you proposed more attractiveness, however, it couldn't contain other information and incomplete.
7. Line 23-26, we revised this sentence.
8. We removed all abbreviations in the abstract.
9. We added their result and conclusion of the previous research in line 81-83.
10. We summarized and rewritten the results of the references in line 104-118.
11. Line 156-162, we added the indices of the higher and lowest genetic diversity.
12. We described the contents of EHM when first appeared in the text.
13. We think the contents in line189-194, line 311-316 and line 327-333 were describe the total results, put it in the results might be better.
14. We gave the reason in the discussion.
15. We deleted the Fig. 7 due to other one reviewer's advice.

16. This paper mainly focus on the genetic relationship of *Pst* in southwestern and northwestern China, the epidemic pathway is other theme, we will analysis this theme by other methods in the next one paper.
17. We removed Liang (2013) in line 543.
18. We deleted the details, and added the corresponding reference.
19. We added the statistics method of Fig. S1 in the materials and methods.
20. In Table S1, we unable to added the longitude and latitude because of too many sampling sites, if we do this, the Table S1 will be too long.
21. We improved and modified all language in the text.

Reviewer #2 (Comments for the Reviewer):

1. In Table S1, in order to present the sampling and grouping relationships more comprehensively, so we put all the information together, there's no dependencies among the different spatial scale subpopulations, if we deal different spatial scales subpopulations to differ Tables, that's lead to too many Tables. In this analysis, the reason why we used single factor to classify isolates is we try to find the *Pst* under grand space in southwestern and northwestern China, whether existed inner correlation between genetic relationship and geography, whatever horizontal and vertical dimensions.
2. It's meaningless to analyze the genetic diversity if a population only contained one subpopulation, otherwise more than one subpopulation. In this research, the Xinjiang population is only one subpopulation on provincial and regional spatial scales. Shaanxi population only counted in Group R as one of two subpopulations. We revised the sentence as: Due to less than three sampling villages or sites, the isolates from Xinjiang province were excluded during the genetic diversity analysis at the provincial and regional levels (Group P and Group R) but included as a part in Group E and L, as well as Shaanxi population was excluded on provincial level but included as a part in Group R, E, and L.

3. In this paper, we used two different vertical spatial scales, varied from relative broader to finer-spatial scale on vertical dimension, that could better to explicit the genetic relationship of *Pst* populations in southwestern and northwestern China.
4. We revised the values of K in Fig.2 ranged from 2 to 9. We used non-parametrical and parametrical models to assess best K values both were 2, so we haven't analyzed the population structure when K ranged from 5 to 10.
5. We enhanced the connection between text and Tables or Figures. Now, line 157 and 159 can be found in Table 2A.
6. The result of genetic differentiation on elevational level revealed the correlation between genetic differentiation and elevation was no standard linear increase, it meant the highest genetic differentiation maybe not existed between subpopulations 0-0.2K and 3.0-3.2K, however, we can find that the relative higher genetic differentiation always occurred between altitude higher than 2,000m and lower than 2,000m. In addition, the result of Group L confirmed this conclusion.
7. We revised the contents of line 50-55, 63-64, 66-67, 92-94, 122-125, and modified the grammar and statements in the whole article.

Minor concerns:

1. We revised the contents in line 104 to 118, and summarized the different previous researches by listing all the results.
2. We deleted the Fig. 7 because of the other reviewer thought too many figures, then we provide the accurate values of R and P in the text.
3. We revised Fig.5 by deleting the values, and the values recorded in Table S3.

Reviewer #3 (Comments for the Author):

1. We added the sampling strategy in the text.
2. It's unavoidable that the samples in different subpopulations with varying the number in population genetics study, the amounts of samples lie on many factors, such as disease incidence,

planting areas, cultivars, sampling time, etc. Milgroom (2015) and Grünwald (2017) recommended the sample sizes for subpopulations and populations are about 10 to 20 individuals and 25 to 30 individuals, respectively (38, 41). In the present study, despite the fewer isolates from Xinjiang and Shaanxi, whereas the total results consistent the previous studies, like as the Shaanxi population close to Gansu, Xinjiang population far away from others, we consider that the findings still provided responsible evidence in genetic relationship between *Pst* populations in southwestern and northwestern. As a precaution, we modified the conclusions about Xinjiang population.

3. We removed Table 1 as supplemental Table S1. In Fig. 3, we remained K=2, and deleted the figures of K=3. For Fig. 7, we deleted this Figure and described the details in the text. We removed “19S” in Fig. 1(A,B,C) and Table 2.

4. We added the meaning of the all abbreviation when they appeared the first time, including in the text, Figure, and Table.

5. Our findings consistent with the previous studies, it is imply the *Pst* populations genetic structures under grand space keep stable, based on the massive *Pst* isolates from the consecutive planting areas to summarize the genetic rules enhance the conclusion, we provided the dependable results.

6. We revised line 157. Gene diversity represent the diversity of each allele, genotypic diversity represents the diversity of MLG (multi-locus genotype) identified by the 12 loci selected.

7. We simplified the methods by cited a reference in line 554-566.

July 8, 2022

Dr. Bingbing Jiang
China Agricultural University
College of Plant Protection
Yuanmingyuan West RD. 2, Haidian District, Beijing
Beijing
China

Re: Spectrum01530-22R1 (The genetic relationship of *Puccinia striiformis* f. sp. *tritici* in southwestern and northwestern China)

Dear Dr. Bingbing Jiang:

Your manuscript has been accepted, and I am forwarding it to the ASM Journals Department for publication. You will be notified when your proofs are ready to be viewed.

Sincerely,

Giuseppe Ianiri
Editor, Microbiology Spectrum

Journals Department
Supplemental Material: Accept
Supplemental Material (figures) FOR publication: Accept